# GOATEX: Geometry & Occlusion-Aware Texturing

**Hyunjin Kim**[*1]     **Kunho Kim**[*2]     **Adam Lee**[3]     **Wonkwang Lee**[†1,4]

[1]KRAFTON AI     [2]NC AI     [3]UC Berkeley     [4]Seoul National University

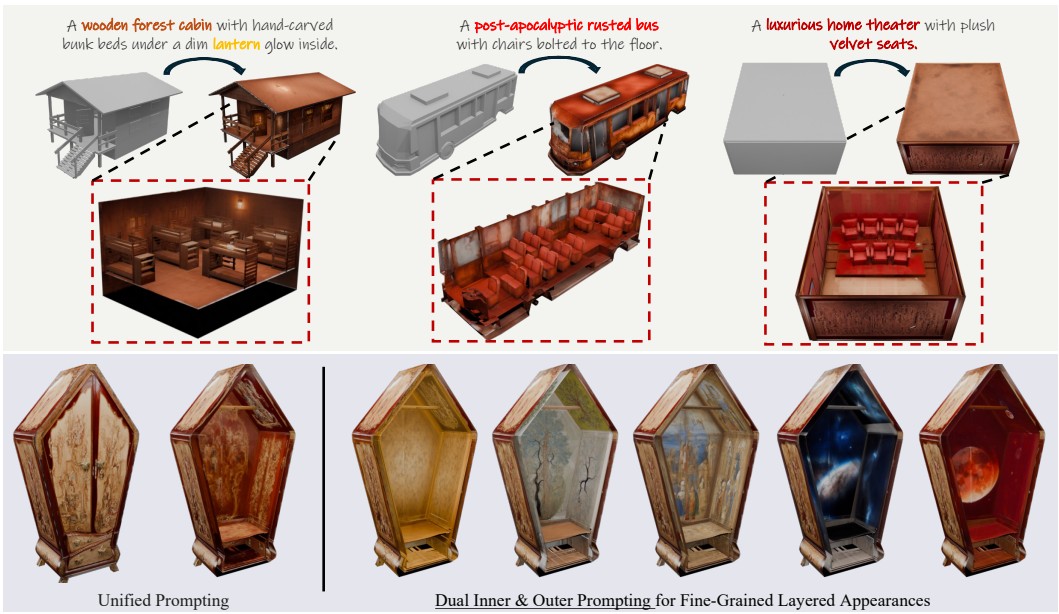

**Figure 1:** We present GOATEX, the first occlusion-aware 3D mesh texturing method designed to synthesize realistic mesh textures for both exterior and occluded interior regions.

## Abstract

We present GOATEX, a diffusion-based method for 3D mesh texturing that generates high-quality textures for both exterior and interior surfaces. While existing methods perform well on visible regions, they inherently lack mechanisms to handle occluded interiors, resulting in incomplete textures and visible seams. To address this, we introduce an occlusion-aware texturing framework based on the concept of hit levels, which quantify the relative depth of mesh faces via multi-view ray casting. This allows us to partition mesh faces into ordered visibility layers, from outermost to innermost. We then apply a two-stage visibility control strategy that progressively reveals interior regions with structural coherence, followed by texturing each layer using a pretrained diffusion model. To seamlessly merge textures obtained across layers, we propose a soft UV-space blending technique that weighs each texture's contribution based on view-dependent visibility confidence. Empirical results demonstrate that GOATEX consistently outperforms existing methods, producing seamless, high-fidelity textures across both visible and occluded surfaces. Unlike prior works, GOATEX operates entirely without costly fine-tuning of a pretrained diffusion model and allows separate prompting for exterior and interior mesh regions, enabling fine-grained control over layered appearances. For more qualitative results, please visit our project page in this link.

[*]Equal contribution     [†]Corresponding Author

39th Conference on Neural Information Processing Systems (NeurIPS 2025).

# 1 Introduction

High-quality mesh textures are critical for creating digital assets across a wide range of applications, including gaming, animation, augmented reality (AR), and virtual reality (VR), as they significantly enhance the believability of virtual environments and improve both user experience and immersion. Importantly, the realism isn't just needed for surfaces that are immediately visible—it also matters for inner details that users might see as they move around, interact with the environment, or change their viewpoint. For example, in architectural visualization, it's not enough to have well-textured house façades; interior features like walls, doors, and furnitures also need detailed textures, since users may examine them closely during a VR walkthrough. Similarly, in vehicle simulation, while the exterior body of a car or bus must appear authentic, interior textures, such as dashboards, seats, and ceiling panels, must also be rendered with high fidelity to support a seamless and immersive experience.

To address these increasing demands for realistic mesh texturing, recent advancements in text-to-image (T2I) diffusion models [42, 37, 40, 35, 20, 46, 34, 28, 29, 14, 43, 10, 9, 49] have played a critical role, advancing image creation by enabling artists and developers to synthesize highly detailed images directly from textual descriptions. Building upon this progress, text-to-texture generation approaches [39, 8, 6, 27, 53, 22, 21, 24, 59, 13, 3, 52] based on 2D image diffusion priors have emerged, aiming to alleviate the labor-intensive process of manual asset creation and make high-fidelity texture generation more accessible. These methods usually operate by unprojecting multi-view rendering of the outer surface onto the mesh's UV map, leveraging 2D diffusion priors to achieve high-quality texture synthesis. Moreover, techniques such as iterative painting [39, 8, 47, 25, 22, 21] and multi-view sampling [6, 27] further enhance view alignments, reducing visible seams and artifacts.

However, texturing interior surfaces remains a significant challenge, as existing methods that rely on multi-view unprojection lack access to occluded or internal geometries, often resulting in untextured regions. Several methods [8, 27] have attempted to address this gap with Voronoi-based filling techniques; however, such methods often result in visible seams and inconsistent surface textures. More recent works [53, 3, 52] have explored UV-space refinement and inpainting by fine-tuning diffusion models. These approaches are capable of texturing internal surface regions; however, they often fail to produce high-quality and plausible interior surfaces due to their reliance on UV space representations, which lack explicit geometric context, and the limited availability of training data that captures the high variability of UV map structures with high-quality interior surface textures.

To this end, we propose GOATEX, the first occlusion-aware mesh texturing method that explicitly targets the underexplored challenge of inner surface texturing. Our key idea is to treat the mesh as a layered structure and progressively reveal surfaces from the outside in, guided by ray-based visibility analysis. We begin by casting multi-view rays to compute *hit levels*, which quantify the relative depth of mesh regions and partition the surface into ordered visibility layers, facilitating a subsequent process of rendering pipeline that progressively exposes occluded geometry. To preserve the overall shape and identity of the object throughout the progression, we introduce a two-stage visibility control strategy that combines residual face clustering with normal flipping and backface culling. This allows new interior surfaces to be exposed without distorting the mesh's global structure. Each layer is then textured independently using MVD module [27] with pretrained depth-conditioned diffusion model [54]. To merge these textures seamlessly, we propose a UV-space blending scheme based on view-dependent visibility weights, which avoids seams and style inconsistencies. Experiments show that GOATEX produces high-quality textures across both visible and occluded regions, while enabling fine-grained control over layered appearance through separate prompts for exterior and interior surfaces.

The contributions of this paper are summarized as follows:

1. To the best of our knowledge, we are the first to introduce and address the task of generating realistic textures for occluded interior surfaces of 3D meshes alongside exterior regions, a practically important yet still underexplored challenge in the 3D mesh texturing literature.
2. We propose GOATEX, a ray-based occlusion-aware framework that textures both exterior and interior regions without requiring tuning of pretrained diffusion models. Our method also supports dual prompting for inner and outer surfaces, enabling fine-grained control over layered structures.
3. User studies and GPT-based evaluations show that GOATEX is strongly preferred over existing methods, achieving state-of-the-art texture quality across both visible and occluded surfaces.

## 2 Related Work

### 2.1 Texture Generation via 2D Diffusion Priors

While significant progress has been made in mesh texturing, earlier methods [31, 30, 17, 32, 41, 44, 33, 7, 45, 16, 5] were often limited by their dependence on scarce high-quality 3D datasets. More recently, the focus has shifted toward zero-shot texture generation using publicly available text-to-image (T2I) diffusion models [40, 35, 54], enabling effective texture synthesis without extensive 3D training data. Many current methods employ a project-and-inpaint strategy [39, 8, 47, 25, 22, 21], synthesizing initial textures from a canonical view and subsequently inpainting missing regions. For instance, TEXTure [39] and Text2Tex [8] incrementally generate seamless textures using depth-conditioned diffusion models guided by trimaps. However, such methods often face cross-view inconsistency due to limited global geometric context. To address this, TexFusion [6] proposes a Sequential Interlaced Multi-view Sampler (SIMS), integrating multi-view appearance cues during denoising for improved consistency. Other approaches [24, 59] optimize UV maps directly through multi-view renderings and score distillation sampling (SDS) [36], although these methods typically require substantial computational resources. SyncMVD [27] further improves coherence through a Multi-View Diffusion module employing view synchronization techniques [2, 26, 23, 50]. Similar to these prior works, our GOATEX leverages powerful 2D diffusion priors but uniquely addresses occluded interior regions. Our method explicitly identifies and progressively textures internal surfaces by employing ray-based visibility analysis to systematically expose and handle inner surfaces.

### 2.2 Geometry-Aware Texture Generation via 3D Mesh-Based Training

Recent advances leverage large-scale 3D datasets, such as Objaverse [12, 11], enabling training of sophisticated texture generation models directly on textured meshes. FlashTex [13] introduces LightControlNet to disentangle lighting effects from surface materials, improving relighting capabilities. Paint3D [53] employs a coarse-to-fine generative framework, initially synthesizing textures with a depth-aware 2D diffusion model and subsequently refining them through dedicated UV Inpainting and UVHD modules. Similarly, TEXGen [52] adopts a hybrid 2D-3D strategy, directly training a specialized UV-space generative model without a coarse-to-fine approach. Geometry-aware approaches like Hunyuan3D-Paint [57] and CLAY [55] utilize mesh geometry conditions, such as normal and position maps, to guide diffusion models that generate multi-view tiled images used for texturing. Specifically, Hunyuan3D-Paint introduces canonical normal and coordinate maps into the diffusion process, incorporating reference and multi-view attention mechanisms, while CLAY synthesizes diffuse, roughness, and metallic maps conditioned on normal maps and reference images. Meta3DTextureGen [3] builds upon these concepts, initially generating multi-view tiled images conditioned on geometry, and further enhancing the texture quality through an additional UV space inpainting network. Despite these advancements, current methods primarily focus on exterior surfaces or, even when generating interior textures in UV space, face difficulties producing plausible interior surfaces due to a lack of high-quality interior texture data.

## 3 GOATEX: Geometry & Occlusion-Aware 3D Mesh Texturing

Texturing the interior surfaces of a 3D mesh poses a unique challenge: these regions are often fully occluded from external viewpoints and receive little to no coverage in conventional rendering-based pipelines. To this end, we propose GOATEX, an occlusion-aware framework for texturing of both exterior and interior mesh regions. A schematic overview of our pipeline is shown in Fig. 2.

### 3.1 Ray-Based Hit Level Assignment for Layered Geometry Decomposition

Our method begins with estimating the relative visibility depth of different regions of the mesh, which allows us to progressively expose and render from exterior to interior surfaces during subsequent texturing stages. This is achieved through two key steps: (1) grouping mesh faces into structurally coherent regions called superfaces, (2) assigning a hit level to each superface based on ray casting.

**Superface Construction.** Directly assigning hit levels to individual faces can result in fragmentation, where adjacent faces belonging to the same planar region are assigned different hit levels due to the mesh's composition of numerous small faces (as in top-left pane of Fig. 2), undermining the generation of semantically consistent and plausible textures in the subsequent stages. To address this,

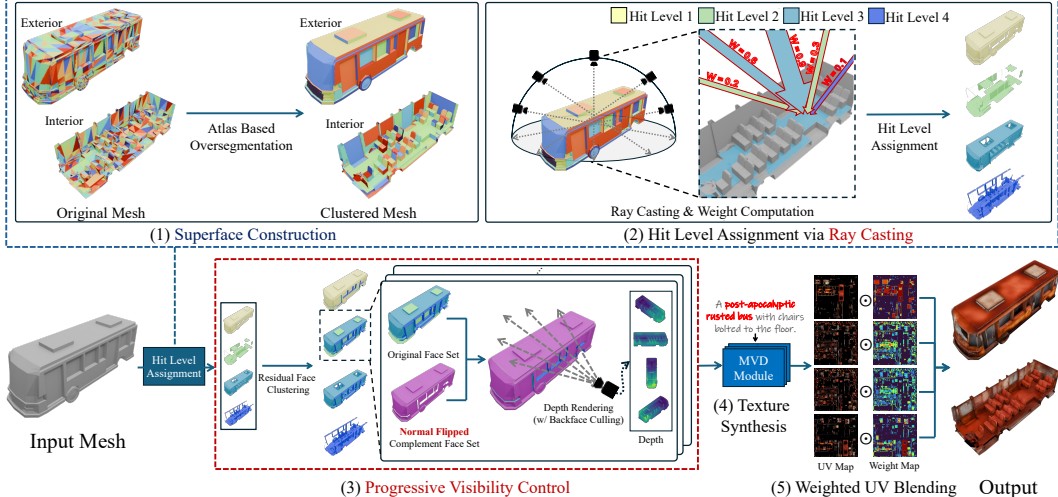

(1) Superface Construction     (2) Hit Level Assignment via Ray Casting

(3) Progressive Visibility Control     (4) Texture Synthesis     (5) Weighted UV Blending    Output

**Figure 2: Overall Pipeline of GOATEX.** (1) Our framework begins with *superface construction*, grouping fine-grained faces into coherent regions. (2) We then assign each superface a *hit level* by casting rays from multiple viewpoints, indicating its relative depth within the mesh (§3.1). (3) Based on these hit levels, we apply a visibility control strategy, combining *residual face clustering* with *normal flipping* and *backface culling*, to progressively reveal interior geometry without disrupting the object's structural integrity (§3.2). (4) For each hit level, we synthesize textures using a depth-conditioned multi-view diffusion (MVD) module [27]. (5) We employ a soft, visibility-weighted *UV-blending* strategy to merge textures across levels, ensuring seamless and coherent appearance (§3.3).

we use Xatlas library [51] to oversegment the mesh into connected, low-curvature regions, referred to as atlases. Each atlas is treated as a superface and serves as the basis for hit level assignment.

**Hit Level Assignment.** Next, we cast rays from multiple viewpoints, recording both their intersection order and directional influence on the surface. These information are then aggregated across all rays to determine the most representative hit level for the superface. Each ray's influence is weighted by the cosine similarity between its direction and the face normal, with more direct (i.e., near $-1$) intersections given greater importance and having a stronger impact on the hit level assignment.

Formally, the influence weight $W(f, k)$ of a ray $r$ with intersection order $k$ on face $f$ is defined as:

$$W(f, k) = \sum_{r \in R_k(f)} \max\left(-n(f) \cdot d(r), 0\right),\tag{1}$$

where $R_k(f)$ is the set of rays intersecting face $f$ with intersection order $k$, $n(f)$ is the normal vector of the face $f$ and $d(r)$ is the direction of the ray $r$. Finally, we aggregate these directional weights to determine the *hit level* $H(SF_i)$, the most dominant intersection order for the superface $SF_i$:

$$H(SF_i) = \arg\max_k \sum_{f \in SF_i} W(f, k).\tag{2}$$

### 3.2 Visibility Control for Structurally Coherent Layered Texturing

Once each face has been assigned a unique hit level based on its visibility depth, a straightforward texturing approach is to render and texture faces independently by hit level, starting from the outermost faces at the lowest hit level and proceeding inward to those at the highest. That is, at each hit level $k$, one could simply construct the set of faces $F_k^{\text{init}} = \bigcup_{H(SF_i)=k} SF_i$ assigned to that level, render their depth maps, and condition a depth-conditioned 2D diffusion model to generate textures for them.

However, this naive strategy suffers from a critical drawback: as illustrated in Fig. 3, the face clusters associated with higher hit levels become increasingly sparse and fragmented. This leads to incomplete and disjointed geometry in the rendered depth maps, which deviate from the coherent structures typically found in natural objects. As a result, the diffusion model receives out-of-distribution (OOD) inputs, impairing its ability to generate plausible textures for interior regions.

We address these issues with a two-stage visibility control strategy: (1) residual face clustering to construct denser, progressively revealed geometry; and (2) normal flipping with backface culling to

preserve global shape. These components together ensure that each depth map preserves both the structural fidelity and contextual coherence necessary for robust texture generation.

**Residual Face Clustering.** To address the sparsity, we redefine the set of faces rendered at each stage by adopting a residual face clustering strategy. That is, rather than rendering only the faces uniquely assigned to hit level $k$, we render the full set of untextured faces remaining from previous levels. The resulting residual face set $F_k^{\text{res}}$ is defined as:

$$F_k^{\text{res}} = F - \bigcup_{i=1}^{k-1} F_i^{\text{init}}, \tag{3}$$

where $F$ is the complete set of mesh faces. Analogous to peeling layers of an onion, this formulation enables the progressive exposure of deeper yet untextured geometry for subsequent texturing.

**Normal Flipping & Backface Culling.** Residual clustering mitigates sparsity but doesn't prevent the monotonic drop in visible faces across deeper hit levels (see Fig. 3, middle). This results in increasingly sparse depth maps that lack structural cues, yielding out-of-distribution inputs for the diffusion model and degraded texture quality.

To this end, we introduce a technique based on *normal flipping* and *backface culling*. At each hit level $k$, we keep all mesh faces but flip the normals of those already textured in earlier levels, turning front-facing surfaces into backfaces relative to current rays, or vice versa. Due to the backface culling, faces whose normals point within $90°$ of the view direction are removed. Thus, flipped surfaces that previously faced the camera (e.g., on the near side) become temporally hidden, as if removed from the scene. Meanwhile, faces once back-facing (e.g., on the far side) may now face the camera and become visible. As a result, this view-dependent behavior reveals untextured interiors, while preserving object structure and enhancing per-view visibility. Formally, rendered faces at level $k$ are:

$$F_k = F_k^{\text{res}} \cup \left( \overline{F} - \overline{F_k^{\text{res}}} \right), \tag{4}$$

where $\overline{F}$ and $\overline{F_k^{\text{res}}}$ denote flipped versions of $F$ and $F_k^{\text{res}}$, respectively.

### 3.3 Weighted UV-Space Blending for Layered Texture Synthesis

Finally, for each hit level $k$, we generate one texture by rendering multi-view depth maps using the visibility-controlled face cluster $F_k$. These depth maps are then used to condition a pretrained texture generation module (MVD module [27]), resulting in a distinct texture map $\text{UV}_k$ for each hit level.

A straightforward way to combine these per-level textures is to simply overwrite or average them across hit levels. However, as illustrated in Fig. 4, such naïve approaches often result in noticeable visual artifacts or the loss of clear boundaries between interior and exterior surfaces.

To address this, we introduce a UV-space texture merging scheme. For each view $v$ at hit level $k$, we compute a UV-space weight $W_k^{(v)}$ based on the absolute cosine similarity between the view direction and the face normal, reflecting how much a given view contributes to each texel in the final texture.

The total visibility weight $W_k$ for each hit level $k$ is computed by aggregating contributions across views and further normalized across the hit levels using a modified softmax function:

$$\overline{W_k} = \frac{e^{W_k} \odot \mathcal{M}k}{\sum_{j=1}^{H} e^{W_j} \odot \mathcal{M}_j}, \; W_k = \sum_v W_k^{(v)}, \tag{5}$$

where $\mathcal{M}_k$ is a binary UV mask indicating valid texels at hit level $k$, and $H$ is the total number of hit levels. The normalized weight $\overline{W_k}$ represents the relative influence of hit level $k$ on each texel.

The final merged texture $\text{UV}_{\text{F}}$ is computed as:

$$\text{UV}_{\text{F}} = \sum_{k=1}^{H} \overline{W_k} \odot \text{UV}_k, \tag{6}$$

where $\text{UV}_k$ is the texture generated at hit level $k$. This soft blending method effectively suppresses artifacts caused by abrupt transitions or simplistic averaging across independently textured regions.

# 4 Experiments

## 4.1 Implementation Details

For all experiments, we use Stable Diffusion 1.5 [40] combined with a depth-based ControlNet [54] to generate multi-view images from input text prompts, where prompts are augmented with view-specific cues [39, 27] to match orientations. Each view is rendered at a resolution of $768{\times}768$, with a corresponding latent resolution of $96{\times}96$. The latent UV texture map has a resolution of $512{\times}512$, and the final RGB UV texture map is generated at $1024{\times}1024$. For visibility and texture synthesis, the maximum hit level is set to 4; we define 16 hemispherical views (8 equatorial at $45°$, 8 elevated at $45°$); Rendering is done with PyTorch3D [38]. Please find Appendix A1.1 for more details.

## 4.2 Experiment Setup

**Dataset.** For evaluation, we curate a diverse set of 139 assets from Objaverse [12] and 87 assets from Objaverse-XL [11], selecting 226 high-quality meshes with detailed interior geometries to assess the performance of our method. Each mesh is normalized to a unit bounding box. To provide diverse and rich prompts that capture both exterior and interior characteristics, we generate mesh captions using GPT-4o. A detailed description of the captioning process is provided in Appendix A2.1.

**Metric.** Mesh texturing literature [8, 6, 27, 53, 52] commonly adopt FID [19], KID [4], and CLIP-based metrics [18, 15], such as CLIP-I and CLIP-T, to evaluate visual quality and text alignment. However, these metrics are ill-suited to our setting for two main reasons. First, ground-truth textures are often unavailable—especially in occluded mesh regions—rendering reference-based metrics like FID/KID and CLIP-I inapplicable. Second, recent work [48, 56, 1] shows that CLIP-based models are sensitive to rendering artifacts and correlate poorly with human judgment in texture assessment.

Therefore, we adopt both a user study and a GPT-based evaluation [48, 56] to assess text alignment and perceptual quality. Specifically, we conduct an A/B preference test, in which participants or a family of GPTs (e.g., GPT-4o-mini, GPT-4o, GPT-4.1, and GPT-o3) view side-by-side renderings of each method's textured mesh, including both exterior and interior views, and select the result that better matches the textual prompt and exhibits higher texture quality. See Appendix A3 for the details.

**Baselines.** We compare our method against publicly available project-and-inpaint approaches (TEXTure [39] and SyncMVD [27]) as well as the methods based on UV inpainting (Paint3D [53] and TEXGen [52]). For TEXGen, since it requires both a mesh and an RGB UV texture map as input, we unwrap the mesh and initialize the UV texture map using the output generated by SyncMVD. An analysis of TEXGen with respect to the input UV map is presented in Appendix A1.2.

## 4.3 Analysis on Visibility Control & UV Texture Merging

**Visibility Control** To assess the effectiveness of our visibility control strategy (§ 3.2), we conduct analysis by ablating residual face clustering and normal flipping. As illustrated in Fig. 3, disabling both components (left pane) results in sparse and fragmented face clusters, particularly evident at hit levels 2 and 4. Introducing residual face clustering alone (middle pane) mitigates the sparsity but still exhibits fragmentation, especially at hit level 4. These limitations result in visible seams in the synthesized textures (e.g., along the edges of a chair) and incomplete reconstruction in occluded areas (e.g., floor surfaces). In contrast, applying both techniques (right pane) produces coherent clusters that reveal interior geometry while maintaining overall object structures in the conditioning views. This reduces artifacts and leads to the generation of semantically meaningful and visually consistent textures, underscoring the importance of both components for high-quality interior surface synthesis.

**UV Texture Blending** We further evaluate the effectiveness of our UV texture merging strategy (§ 3.3). In our visibility control scheme, the same mesh region can be textured across multiple hit levels, each with varying visibility confidence depending on the angle between the viewing ray and the surface normal. This overlap introduces challenges in merging textures from different levels. As shown in Fig. 4, naive strategies, such as direct overwriting or uniform averaging, fail to account for these confidence variations. They either indiscriminately replace textures or blend them without regard to visibility, leading to noticeable artifacts and loss of structural detail, particularly at inner and

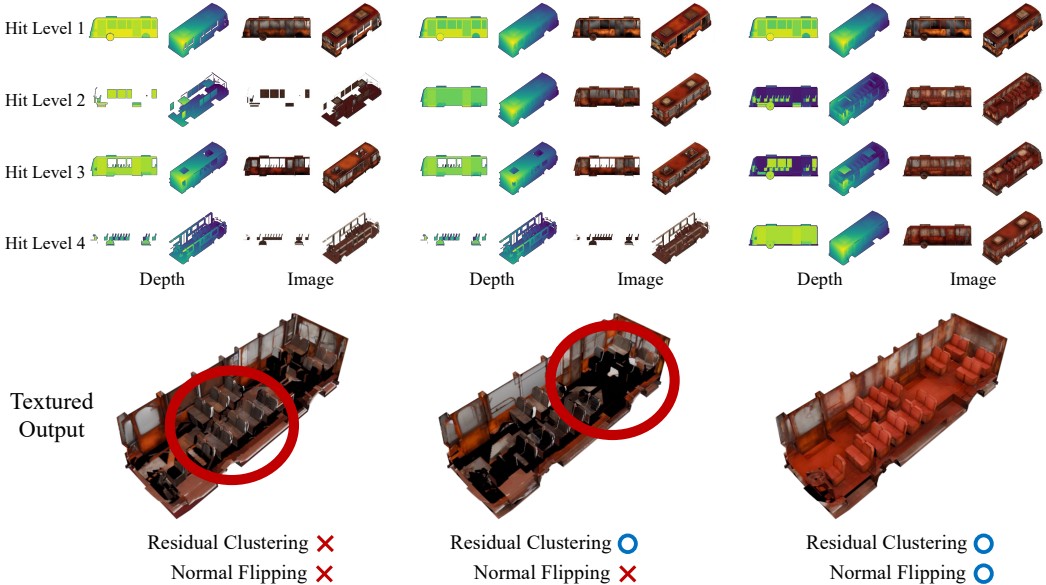

**Figure 3:** Analysis on our two visibility control techniques, residual face clustering and normal flipping. (Left) Removing both leads to sparse, fragmented face clusters and causes visible seams (e.g., around chair edges). (Middle) Residual clustering improves cluster density but still suffers from fragmentation at hit level 4, suffering from incomplete textures (e.g., floor regions). (Right) In contrast, combining both techniques results in improved visibility segmentation, effectively revealing interior structures while preserving the overall shape and structural coherence of the object.

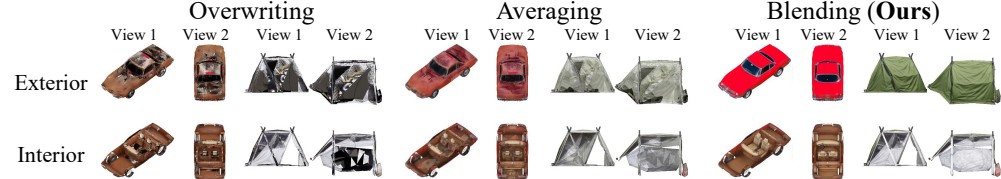

**Figure 4:** Analysis on UV texture merging techniques. Compared to naive approaches, our blending strategy preserves clear structural boundaries and significantly improves texture quality.

outer boundaries. In contrast, our weighted blending approach leverages hit-level-specific visibility confidence to guide the merging process. This enables smooth and fine-grained transitions between layers, significantly reducing artifacts and enhancing overall visual quality.

## 4.4 Qualitative Comparison with Baselines

We qualitatively compare our method against existing approaches, as shown in Fig. 5. TEXTure and SyncMVD struggle to synthesize plausible interior textures due to their reliance on view-based generation followed by unprojection. This paradigm inherently lacks access to occluded geometry, leading to heuristic-based filling (e.g., Voronoi-based extrapolation) that results in simplistic, inconsistent textures and visible seams. In contrast, UV-space generation methods such as Paint3D and TEXGen show improved performance in interior regions by operating directly in UV space. However, they still exhibit limitations, often producing low-frequency textures like flat colors or repetitive patterns, because they are unable to differentiate between interior and exterior surfaces within the UV map. This design limitation reduces semantic richness and structural coherence in occluded areas.

Among all methods, GOATEX produces the most visually appealing and semantically coherent textures across both exterior and interior regions. This is achieved through its ray-based layered surface decomposition, which explicitly exposes occluded geometry, followed by visibility control strategies that progressively reveal hidden surfaces. These components provide precise geometric context, enabling the diffusion model to generate rich and plausible textures. Unlike baselines that

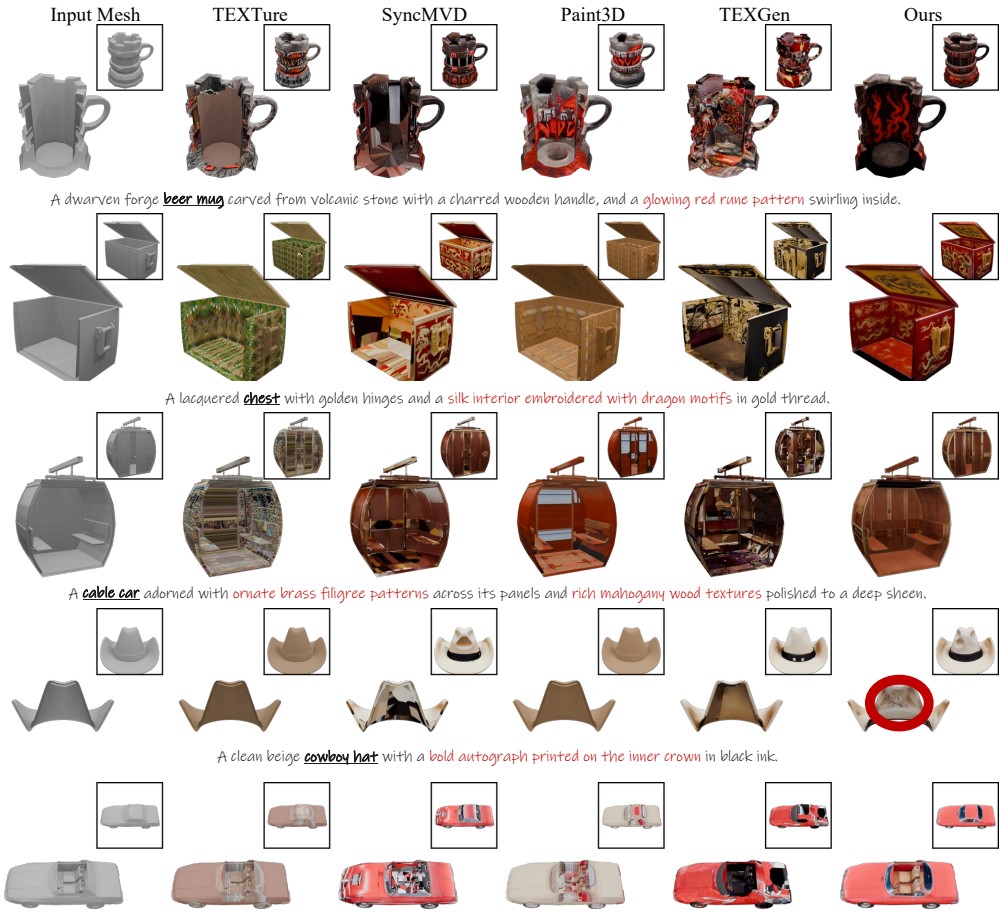

**Figure 5:** Qualitative comparisons. Our GOATEX significantly outperforms all the baselines in interior surface texturing, while maintaining competitive in texturing exterior regions (boxed area at the top right of each object). This balanced approach makes GOATEX a robust solution for both visible exterior and occluded interior surface texturing.

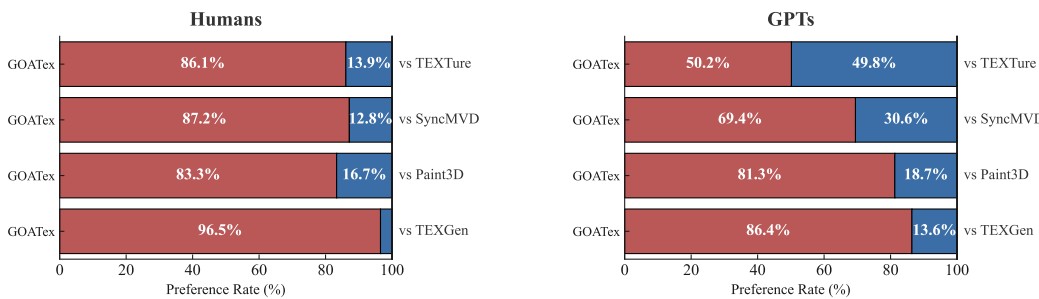

**Figure 6:** Preference rates of our method over the baselines, as judged by human raters or GPTs.

rely on heuristics or operate blindly in UV space, GOATEX performs structurally informed texturing of all surfaces, resulting in consistent and high-quality outputs across the entire mesh.

## 4.5 Quantitative Comparison with Baselines

Next, we present quantitative comparison results in Fig. 6. GOATEX achieves strong and consistent preference from human raters across all comparisons. On the other hand, GPT-based evaluations rate

its relative advantage lower when compared to TEXTure and SyncMVD. We speculate that this is because GPTs tend to favor results with smoother interiors—such as those produced by TEXTure and SyncMVD, which either leave the interior unpainted or extend exterior textures inward through texture bleeding or heuristic Voronoi-based filling—over methods that explicitly paint interior regions, including Paint3D, TEXGen, and our GOATex. This suggests that GPTs may place greater emphasis on surface smoothness than on interior completeness and consistency.

To further quantify the agreement between human and GPT-based evaluations, we measured two types of metrics: (1) correlation using Pearson's $r$ ($-1 \leq r \leq 1$, where $r = 1$ indicates perfect correlation), and (2) inter-rater agreement using Cohen's $\kappa$ ($\kappa = 1$ indicates perfect agreement). Pearson correlations between GPT-based and human ratings were: GPT-4o-mini: 0.22, GPT-4o: 0.31, GPT-4.1: 0.43, and GPT-o3: 0.34. Cohen's $\kappa$ values (averaged over $\kappa > 0$) were: GPT-to-GPT: 0.54, user-to-user: 0.31, and user-to-GPT: 0.27.

The results reveal two key trends: (1) more advanced GPT models (e.g., GPT-4.1 and GPT-o3) exhibit stronger alignment with human raters, and (2) GPT models demonstrate higher internal consistency than human raters themselves. Interestingly, in 17 individual evaluation cases, user-to-GPT agreement exceeded $\kappa = 0.5$, and in two cases, GPT-o3 achieved perfect agreement ($\kappa = 1.0$) with a human rater, indicating that GPT-based judgments can closely reflect human perception in specific contexts.

## 4.6 Ablation Study

To better assess the contribution of each component in our occlusion-aware texture generation framework, we conduct an ablation study by progressively adding modules to the baseline texturing method (SyncMVD [27]). Following the same protocol as the quantitative evaluation in Sec. 4.5, each ablated variant is compared against SyncMVD via A/B preference tests using GPT-based evaluators. The overall win rates of the ablated configurations over the baseline are summarized in Tab. 1.

The results show consistent performance gains as the proposed components are progressively incorporated. A slight drop is observed when residual face clustering is applied without normal flipping and backface culling. This behavior is expected, as residual clustering alone cannot fully resolve occlusions. In certain cases, geometrically adjacent faces may be grouped into the same hit level even when one fully occludes the other from all external viewpoints. Consequently, the occluded face cannot be textured during its designated rendering stage and is subsequently excluded once that level completes, leading to missing interior details. In contrast, incorporating normal flipping and backface culling effectively suppresses already-textured outer surfaces while exposing previously hidden inner faces, ensuring that occluded regions become visible and correctly textured in subsequent stages.

**Table 1:** Ablation study of the proposed framework. Each component is cumulatively added on top of the baseline (SyncMVD [27]), and evaluated via A/B preference tests using multiple GPT-based evaluators. Scores denote the win rate (%) over the baseline, showing consistent improvements as components are integrated.

| Method | 4o-mini | 4o | 4.1 | o3 | Avg. |
|---|---|---|---|---|---|
| Baseline (SyncMVD) [27] | - | - | - | - | - |
| ✚ Hit Level Assignment | 82.50 | 66.67 | 75.00 | 77.50 | 75.68 |
| ✚ Superface Construction | 84.62 | 70.00 | 75.86 | 89.74 | 80.27 |
| ✚ Soft UV Merging | 79.49 | 82.05 | **90.00** | 95.00 | 86.49 |
| ✚ Residual Face Clustering | 77.50 | 72.50 | 88.89 | 84.84 | 81.17 |
| ✚ Normal Flipping & Backface Culling (**Ours**) | **86.84** | **92.31** | 86.67 | **97.50** | **91.16** |

## 4.7 Separate Prompt Conditioning for Inner and Outer Surfaces

A key advantage of GOATEX is its ability to support distinct prompts for exterior and interior mesh regions: Since textures are synthesized independently for each hit level via our text-guided MVD module, the textual prompt can be specified separately for each level. We demonstrate this capability in Fig. 1 and Fig. 7, where we assign one prompt to the outermost layer (hit level 1) to control the exterior appearance, and another to deeper layers (hit levels $\geq 2$) to govern interior textures. As illustrated, GOATEX produces stylistically distinct and contextually appropriate textures for

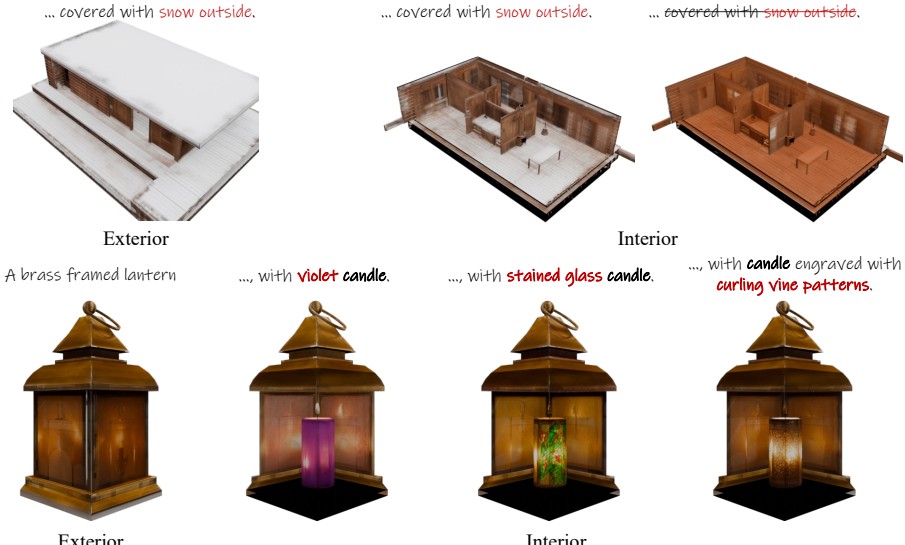

**Figure 7:** Dual prompting for fine-grained interior regions texturing. Unlike previous work, our framework naturally supports distinct text prompt conditioning for synthesizing semantically coherent and stylistically diverse textures across exterior and interior mesh regions.

interior regions, reflecting the semantics of the specified prompt. This feature introduces a new axis of controllability, empowering users to induce rich stylistic variation across surface layers through prompt design. It broadens the expressive potential of 3D assets and enables more diverse, imaginative 3D scene generation.

## 5 Discussion

We introduced GOATEX, a novel diffusion-based framework for 3D mesh texturing that addresses the challenge of generating high-quality textures for both exterior and occluded interior surfaces. By leveraging a ray-based visibility analysis to decompose the mesh into ordered layers, GOATEX enables progressive, occlusion-aware texturing from the outside in. Our two-stage visibility control mechanism ensures structural consistency during layer exposure, while the use of pretrained diffusion models ensures both quality and flexibility. A key strength of our approach is its support for separate prompting of inner and outer surfaces, enabling fine-grained stylistic control and expanding the design space for 3D content creation. Experimental results demonstrate that GOATEX outperforms existing methods in both texture fidelity and semantic coherence, particularly in regions that were previously difficult to access or stylize. We believe that GOATEX opens up new possibilities for controllable, high-fidelity mesh texturing, especially in applications requiring immersive and detailed environments. Future work may explore extending our framework to dynamic scenes, integrating material properties beyond texture, or further improving blending strategies for more complex topologies.

Despite the potential, our method has one primary limitation. Our current hit-level assignment is determined purely by geometric visibility (i.e., ray-intersection depth) and does not explicitly account for semantic coherence. Consequently, in complex geometries such as objects with thin openings or nested cavities, semantically unified regions may be divided across multiple hit levels, which can in turn cause minor texture discontinuities at their boundaries. In practice, however, our residual face clustering, view-dependent normal flipping, and soft UV-space blending effectively mitigate most of these artifacts, producing semantically plausible and visually coherent textures. Nevertheless, incorporating semantic-aware refinement into the hit-level assignment, e.g., grouping superfaces belonging to the same semantic volume using pretrained part-segmentation models, could further improve cross-region consistency. Future work may also explore integrating soft blending directly into the denoising process to enable simultaneous multi-view and multi-hit-level texturing for tighter cross-level coherence.

## Acknowledgments and Disclosure of Funding

Kunho Kim was supported by Institute for Information & communications Technology Promotion(IITP) grant funded by the Korea government(MSIT) (RS-2024-00398115, Research on the reliability and coherence of outcomes produced by Generative AI). Kunho Kim would like to thank Chanran Kim (Pseudo Lab) for the GPU support. Hyunjin Kim thanks Gihyun Kwon, Joo Young Choi, Juhyeong Seon for their helpful discussions.

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

# Appendix

In the appendix, we present additional implementation details (Sec. A1), the dataset gathering (Sec. A2.1) and captioning (Sec. A2.2) process, the evaluation setup (Sec. A3), a more detailed explanation of our method (Sec. A5), limitations (Sec. A6), broader impacts (Sec. A7), and additional qualitative results (Sec. A8). For information on the input text prompts not shown in the main paper and further results not included here, please refer to the project page link.

## A1 Additional Implementation Details

### A1.1 Detailed Experimental Setup

The experiment was conducted using a single RTX A6000 GPU, requiring 12GB of memory per inference. The inference time depends on the number of mesh faces and is calculated as (number of hit levels) × (inference time of the texture synthesis model). For hit level assignment, a total of 17 cameras were used: the original camera views from texture generation plus an additional top-view camera. Ray casting for hit level assignment was performed at a resolution of 1536×1536 using the Open3D [58] library. Since we utilize a pretrained Depth ControlNet [54] without additional training, no further training or training dataset is required.

### A1.2 Implementation Details for Baselines

TEXGen [52] need a initial UV map that is completely aligned with input geometry. However, since the ground-truth UV map is unavailable for the mesh, we performed initialization through the three methods illustrated in Fig. A8 to execute TEXGen. All experiments were conducted using the SyncMVD [27] output UV map—which produced the most plausible results—as the input UV map for TEXGen.

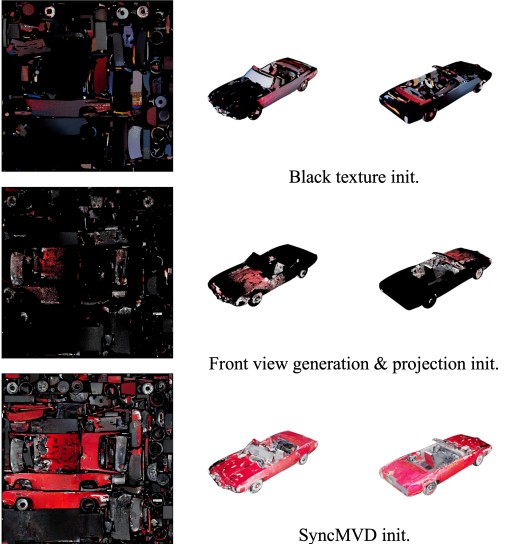

Black texture init.

Front view generation & projection init.

SyncMVD init.

**Figure A8:** TEXGen input UV map ablation.

## A2 Dataset

### A2.1 Dataset Gathering

We curate our dataset from Objaverse [12] and Objaverse-XL [11], selecting 226 high-quality meshes with detailed interior geometries to evaluate our method. To ensure diversity, we focus on 12 object categories that are likely to contain interior structures: box, bucket, bus, cabinet, car, drawer, house, lamp, room, shelf, tent, and truck.

Our primary selection criterion targets objects exhibiting realistic interior geometry at hit levels above 1. This includes two types of objects: (1) those with partially visible interior geometry due to occlusions in canonical views but are not fully enclosed (e.g., tents, igloos, half-open boxes), and (2) topologically closed objects whose interior geometries are entirely enclosed and occluded from all external views (e.g., houses, cars, buses).

We filter candidate meshes using metadata, searching for relevant keywords in descriptions and retaining only those with hit levels greater than 4. Each selected mesh is then manually inspected to ensure it contains realistic and non-trivially simplified interior geometry.

### A2.2 Captioning

To generate rich and diverse captions that capture both the exterior and interior features of 3D objects, we follow a two-step process involving key visual element identification and prompt generation.

**Key Visual Elements Identification.** We begin by rendering each mesh from a fixed three-quarter viewpoint at four different hit levels, progressively revealing surfaces from the outermost to the innermost layers. We then task GPT-4o to analyze these multi-layered renderings to identify salient visual elements and structural features specific to each depth level. See Figure A9 for the detailed system prompt for this task.

**Prompt Generation.** Using the visual features extracted in the previous step, GPT-4o then generates 10 diverse text prompts for each mesh. These captions are crafted to explicitly describe both the external form and internal structure of the object and are later used as conditioning inputs for text-to-texture generation. For the system prompts used in this step, refer to Figure A10.

**Prompt Template for Identifying Key Visual Elements and Structures**

**[System Instruction]**
You are given an object mesh file described through two modalities:
1. A caption summarizing the object.
2. Up to four rendering images that sequentially reveal the structure of the mesh.
    - The first image shows the outermost surface of the object.
    - The following images (`image2` to `image4`) gradually reveal interior details of the mesh.

Your task is to identify only the most prominent, large-scale objects or structures within the mesh, using both the caption and the rendering images.
- Ignore small or minor details.
- Focus only on components that are visually dominant or structurally significant.
- Include any objects or components mentioned in the caption, but remove descriptive modifiers (e.g., `wooden cabin` → `cabin`).

**Output Format**
Your output should be in the following JSON format:

```
{
    "objects": [/* list of object/component names as strings */]
}
```

**Examples**
**Input:**
Caption: Jungle tent
Images: <image1>, <image2>, <image3>, <image4>
**Output:**

```
{
    "objects": ["tent", "candle", "sofa"]
}
```

**Input:**
Caption: Vintage car
Images: <image1>, <image2>, <image3>, <image4>
**Output:**

```
{
    "objects": ["car", "seats", "handle", "dashboard"]
}
```

**Input:**
Caption: A hollow pumpkin head
Images: <image1>, <image2>, <image3>, <image4>
**Output:**

```
{
    "objects": ["pumpkin head"]
}
```

**[User Prompt]**
This is the caption: {caption}.
These are the rendering images: {image1} {image2} {image3} {image4}.
Now, please identify the objects or components in the mesh.

Figure A9: Prompt template used for identifying key visual elements and structures in a mesh.

**Prompt Template for Caption Generation from Object Mesh Files**

**[System Instruction]**
You are given:
- A base caption that describes an object mesh.
- A list of component names (parts of the mesh).

Your task is to generate 10 unique and diverse one-sentence captions that describe both the outer and inner aspects of the mesh, focusing on the material, texture, style, and pattern of the components.

**Requirements:**
- Do NOT describe the overall shape or structure of the object or any of its components.
- Do NOT mention colors directly.
- Use ambiguous or stylistic terms that imply visual variety (e.g., "glossy", "worn", "textured", "transparent", etc.).
- Captions must reflect different styles, materials, or vibes — avoid repetition.
- Each caption should reference the object's components, using the provided component list.
- The tone and style of the caption can differ from the original (e.g., modern, fantasy, sci-fi, surreal, etc.).
- Each caption must be a single sentence.
- Output must be in JSON format with the key "ten_captions".

**Example 1:**
**Input:**
```
{
  "caption": "Red retro bus on a sunny day",
  "objects": ["bus", "seats", "windows"]
}
```
**Output:**
```
{
  "ten_captions": [
    "A modern city bus interior filled with molded plastic chairs ...",
    "A vintage tour bus with upholstered velvet chairs arranged in ...",
    "A futuristic electric bus with sleek metallic chairs glowing ...",
    "A school bus interior lined with simple padded chairs and scratched ...",
    "A fantasy woodland bus with carved wooden chairs and vines wrapping ...",
    "A luxurious travel bus with reclining leather chairs and soft ambient ...",
    "A post-apocalyptic bus interior with mismatched salvaged chairs ...",
    "A steampunk bus with brass-framed chairs covered in worn leather...",
    "A magical flying bus with floating chairs made of translucent crystal",
    "A retro sci-fi bus interior with bubble-shaped plastic chairs and ..."
  ]
}
```

**[User Prompt]**
This is the original caption: {caption}.
These are the list of componetns: {components}.
Now, please generate 10 unique and diverse one-sentence captions that describe both the outer and inner aspects of the mesh, focusing on the material, texture, style, and pattern of the components.

Figure A10: Prompt template for generating stylistic captions from object mesh data.

## A3 Evaluation Setup

As described in Sec. 4.4 of the main paper, we report preference statistics based on responses from user study and GPTs. Below, we provide additional details about the user study setup and protocol. Specifically, following the visualization style used on our project page, we rendered for each method:

- one GIF showing the exterior of the object, and
- one GIF showing the interior via a cut-away or sliced view.

These side-by-side visualizations were shown to participants or GPTs, providing one exterior and one interior view per method per object. We report results from 16 human raters who passed a vigilance test. The screen example of the user study is illustrated in Fig. A11.

To enable a more scalable preference analysis, we additionally conduct an automated evaluation using a family of GPTs, including GPT-4o-mini, GPT-4o, GPT-4.1, and GPT-o3. For this study, we sample a total of 400 tasks. To eliminate potential bias due to ordering, each comparison is evaluated in both A/B and B/A presentation orders. Moreover, if the two judgments from opposite orders disagree, the evaluation is repeated up to 10 times until a consistent decision is reached. Tasks that still yield inconsistent results after 10 repetitions are excluded from the final analysis. The detailed system prompt used for this evaluation is shown in Fig. A12.

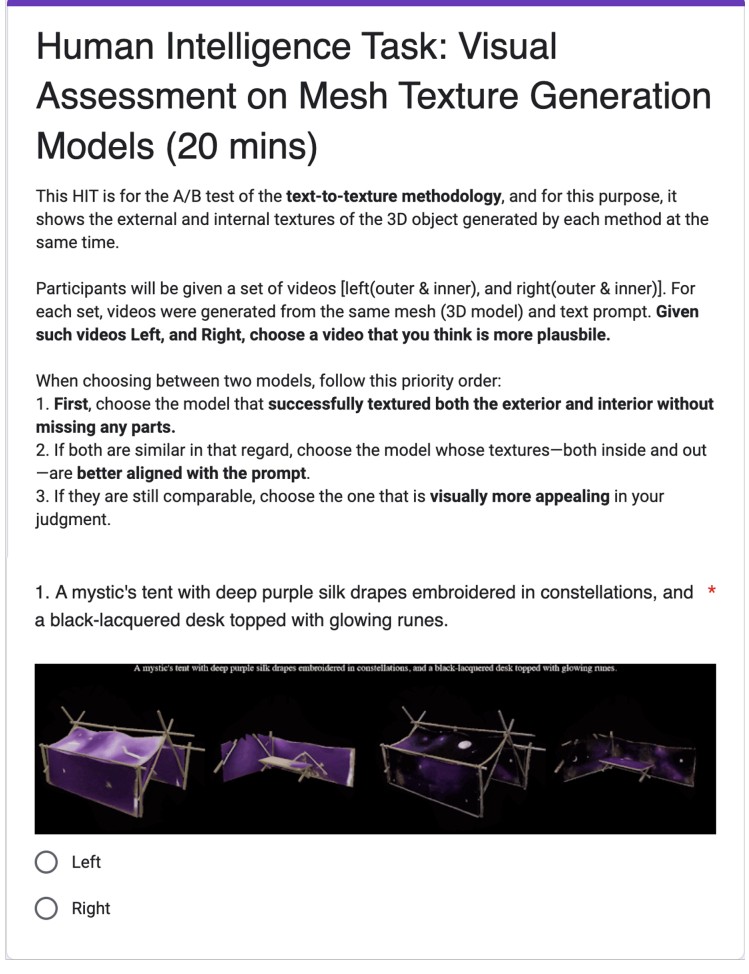

**Figure A11: Screen example of user study**. The example depicts evaluation guidelines and a question sample used in the user study.

**Instruction for Evaluating Textured Mesh**

**[System Instruction]**
You are given a text caption and two sets of rendered images generated by two anonymized text-to-texture models (Model A and Model B). Each model attempts to synthesize realistic textures for both the exterior and interior of a 3D object based on the given caption.

**Each model provides:**
- 3 external renderings, showing the textured outer surface of the object from different viewpoints.
- 3 internal renderings, showing the textured interior structure, such as cutaways or cross-sections.

**Your task is to:**
- Carefully read the text caption, which may describe details about the object's outer appearance, its internal textures or structure, or both.
- Evaluate how well each model's textures match the caption:
    - Use the external renderings to judge how well the outer textures reflect the description.
    - Use the internal renderings to judge how well the interior textures or structures align with the caption.
- Compare the two models holistically and determine which model (A or B) better captured the textures described in the caption, for both external and internal parts of the object.

**Important Notes:**
- The model names 'A' and 'B' are anonymized. You must avoid any bias based on the label itself.
- Your judgment should be based only on the alignment between the caption and the textures shown in the renderings.

**Output Format**
Return the name of the model that performed better: A or B.

**[User Prompt]**
This is the prompt: {prompt}.
Here are rendering images for the Model A. Outer: {image1} {image2} {image3}. Inner: {image4} {image5} {image6}
Here are rendering images for the Model B. Outer: {image1} {image2} {image3}. Inner: {image4} {image5} {image6}
Now, please choose model A or model B based on the caption and the images provided.

Figure A12: Instruction for evaluating GPT-based A/B test.

## A4 Runtime Analysis

We systematically analyzed the runtime of our pipeline across meshes with increasing face counts by progressively subdividing mesh faces of five representative assets. For each mesh, we measured the number of faces and superfaces, time spent on superface construction, hit-level assignment, and MVD-based rendering & synthesis. The table A2 shows the average runtime across the assets. Note that hit-level assignment is reusable when generating multiple variants, so we can preprocess the mesh before texturing.

**Table A2:** Runtime analysis of each module according to the number of faces.

| # Faces | # Superfaces | Superface Construction (s) | Hit-Level Assignment (s) | Texturing (s) |
|---|---|---|---|---|
| 5k | 196.0 | 0.23 | 259.57 | 117.93 |
| 10k | 221.2 | 0.46 | 295.38 | 121.80 |
| 20k | 247.6 | 0.86 | 314.16 | 130.42 |
| 40k | 433.0 | 1.87 | 394.67 | 146.45 |
| 80k | 586.0 | 3.99 | 539.68 | 197.29 |
| 160k | 2033.2 | 18.59 | 942.36 | 549.83 |
| 320k | 5418.0 | 57.76 | 2563.35 | 2769.22 |

## A5 More Detailed Method Explanation

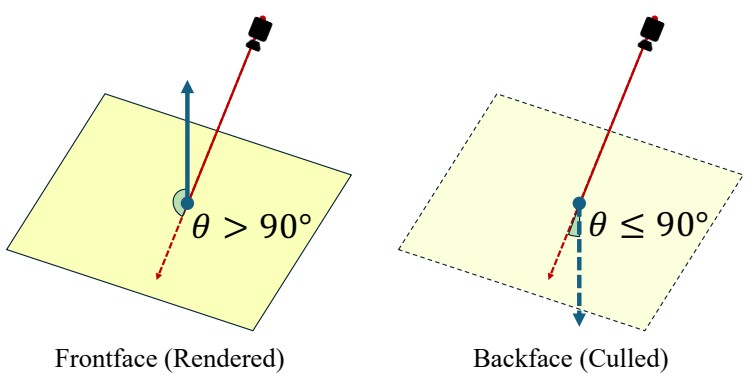

Frontface (Rendered)  Backface (Culled)

**Figure A13:** Rendering with backface culling.

**Mesh & Texture Representation.** In standard mesh representations, each face is a single-sided surface that does not distinguish between its front and back; both sides appear visually identical. This means that any texture or color applied to one side of a face is also visible from the opposite side. Consequently, a mesh face does not have separate visual properties for the inside and outside; regardless of the viewing direction, the appearance remains the same.

To allow for different appearances on the two sides of a surface, such as when modeling an object with distinct exterior and interior textures, it is common to place two faces very close together, back to back. The outer face, slightly offset toward the exterior, is meant to be visible from outside the object, while the inner face is offset inward and intended to be seen from the inside. This setup allows each side to use its own texture and material properties. For example to represent both the painted exterior of a bus and its interior wall surface, the mesh includes one face for the outer shell and another for the inner wall. Each face can be textured independently to reflect its visual role.

**Rendering with Normal Flipping & Backface Culling.** During rendering with backface culling, faces whose normal vectors point away from the camera are not rendered. More precisely, as illustrated in Fig A13, when the angle $\theta$ between the viewing ray and the face normal falls within the

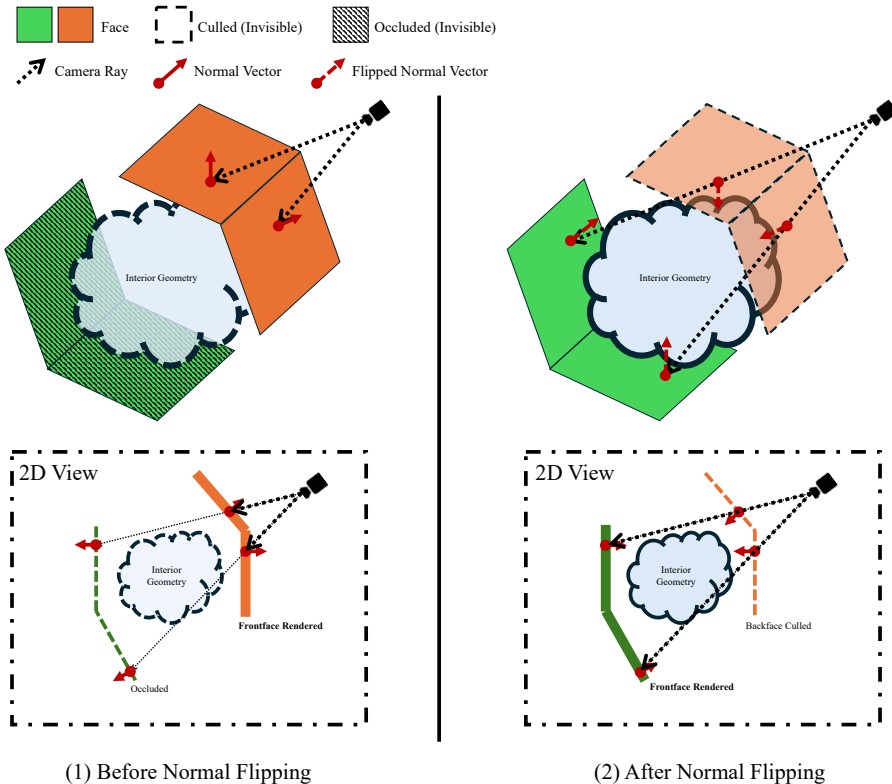

(1) Before Normal Flipping         (2) After Normal Flipping

**Figure A14: Progress of Normal Flipping & Backface Culling**. By flipping the normals and rendering with backface culling, we can successfully view the interior geometry while keeping the object boundaries visible, thus preserving the overall object identity and structure.

range $-90° < \theta < 90°$, the face is considered a back face and is culled from the rendering process. As a result, flipping the normal of a face that was previously visible causes it to become hidden, effectively simulating the removal of the face from view. We use this property to progressively reveal the internal geometries while preserving the object's overall shape and structures during subsequent rendering stages.

We further illustrate this process in Fig. A14, which depicts the current outermost layer of face clusters at rendering stage for hit level $k$ (i.e., $F_k$), divided into two regions colored in orange and green, along with the occluded internal geometry shown in blue. For simplicity, we assume a single fixed camera viewpoint. Initially, the normals of $F_k$ are oriented outward, making only the orange region—being front-facing relative to the camera—visible and rendered, while the internal geometry remains hidden. In the next step, the normals of $F_k$ are flipped inward. As a result, the orange region becomes back-facing with respect to the camera and is culled during rendering, thereby revealing portions of the previously hidden internal geometry. Simultaneously, the green region of $F_k$, which was initially back-facing and thus invisible, now has its normal oriented toward the camera. This allows the green region to become visible, unless it remains occluded by inner layers. This progressive rendering approach preserves the continuity of the overall geometry while revealing deeper layers, maintaining the object's global structure, boundary, and identity.

One potential concern arises when some internal mesh faces have normals directed inward, toward the object center, rather than outward. In such cases, if these inward-facing normals are flipped outward to make the faces visible from the camera, they can occlude deeper surfaces within the object, preventing access to further interior geometry (Fig. A15 (1)). However, this issue does not arise in practice due to the way hit levels are assigned. Specifically, hit levels are computed based on weights derived from Eq. 1, which use the negative cosine of the angle between face normals and ray directions (see Fig. A16). As a result, faces that are originally back-facing receive hit levels in the reverse order of intersection along the ray path, as illustrated in Case 2 of Fig. A15. This property ensures that when normal flipping is combined with backface culling, a valid hit level always exists that exposes each layer of interior geometry. Consequently, both exterior and interior surfaces can be progressively revealed and fully textured (Fig. A15 (2)).

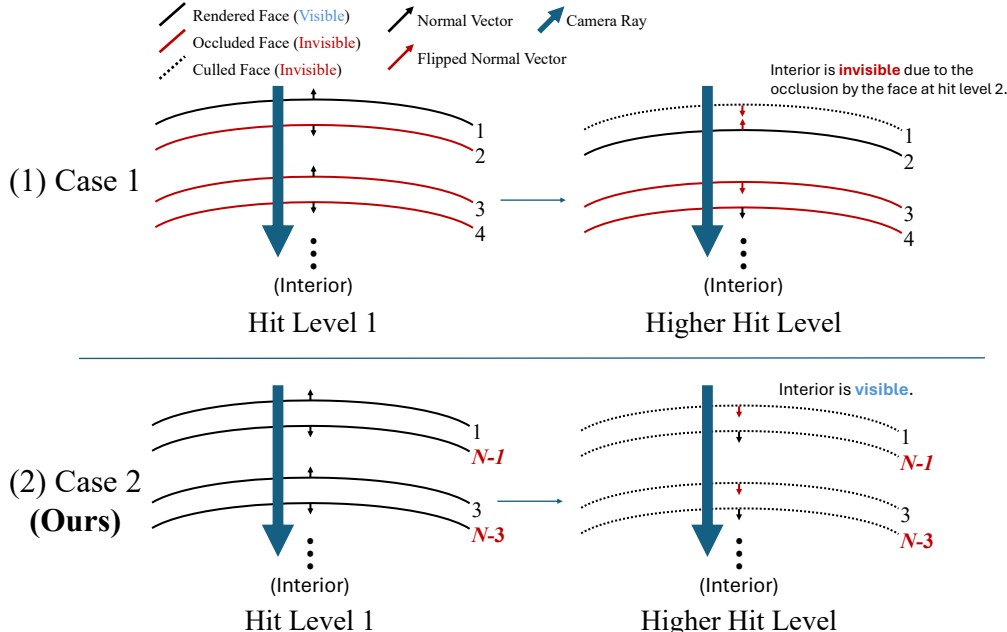

**Figure A15: Visibility According to Face Hit Level Assignment.** When face normals are complexly oriented and hit levels are assigned sequentially from the exterior inward (as in Case 1), applying normal flipping and backface culling during rendering results in occlusion caused by faces at Hit Level 2, preventing visibility into interior regions. Conversely, in Case 2, where faces closer to the camera ray are assigned lower hit levels if they are frontfaces and higher hit levels if they are backfaces, complete visibility of interior regions can be achieved.

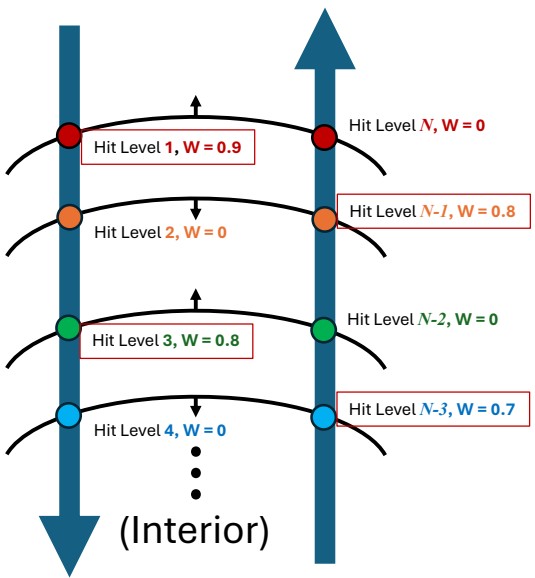

**Figure A16: Hit Level Assignment Based on Normal Direction.** For faces close to camera rays coming from outside the object (left side), backfaces—whose normals align similarly with ray directions—are assigned a weight of zero, resulting in no hit level assignment. In contrast, rays approaching from the opposite direction (right side) encounter these same faces as frontfaces, thereby receiving hit level assignments. However, due to their distance from the camera, these faces are assigned higher hit levels.

## A6  Limitations

Our approach has a few limitations. While GOATEX is effective at generating high-quality interior textures, its reliance on the SD 1.5 backbone can occasionally result in suboptimal adherence to input text prompts. In future work, we plan to integrate more lightweight yet more capable pretrained diffusion models to enhance prompt fidelity and further improve texture quality.

Our current hit-level assignment is determined purely by geometric visibility (i.e., ray-intersection depth) and does not explicitly account for semantic coherence. Consequently, in complex geometries such as objects with thin openings or nested cavities, semantically unified regions may be divided across multiple hit levels, which can in turn cause minor texture discontinuities at their boundaries. In practice, however, our residual face clustering, view-dependent normal flipping, and soft UV-space blending effectively mitigate most of these artifacts, producing semantically plausible and visually coherent textures. Nevertheless, incorporating semantic-aware refinement into the hit-level assignment, e.g., grouping superfaces belonging to the same semantic volume using pretrained part-segmentation models, could further improve cross-region consistency. Future work may also explore integrating soft blending directly into the denoising process to enable simultaneous multi-view and multi-hit-level texturing for tighter cross-level coherence.

## A7  Broader Impacts

**Potential Positive Societal Impacts.**  It will be possible to further accelerate the existing 3D asset creation pipeline and generate a wider variety of assets. With these assets, we can expect to build even more realistic AR/VR environments. As a result, people will be able to enjoy a broader range of experiences.

**Potential Negative Societal Impacts.**  Texture generation models present potential risks, such as the synthesis of deepfakes, textures resembling copyrighted content, or biased and discriminatory textured meshes. Future work should focus on developing robust mechanisms to mitigate these risks and enforce safeguards that prevent the generation of harmful or unethical outputs.

## A8 Additional Qualitative Results

More qualitative comparisons can be found in Fig. A17, and additional qualitative results are shown in Fig. A18. For a more detailed presentation—including videos of the results and further outcomes—please see this link.

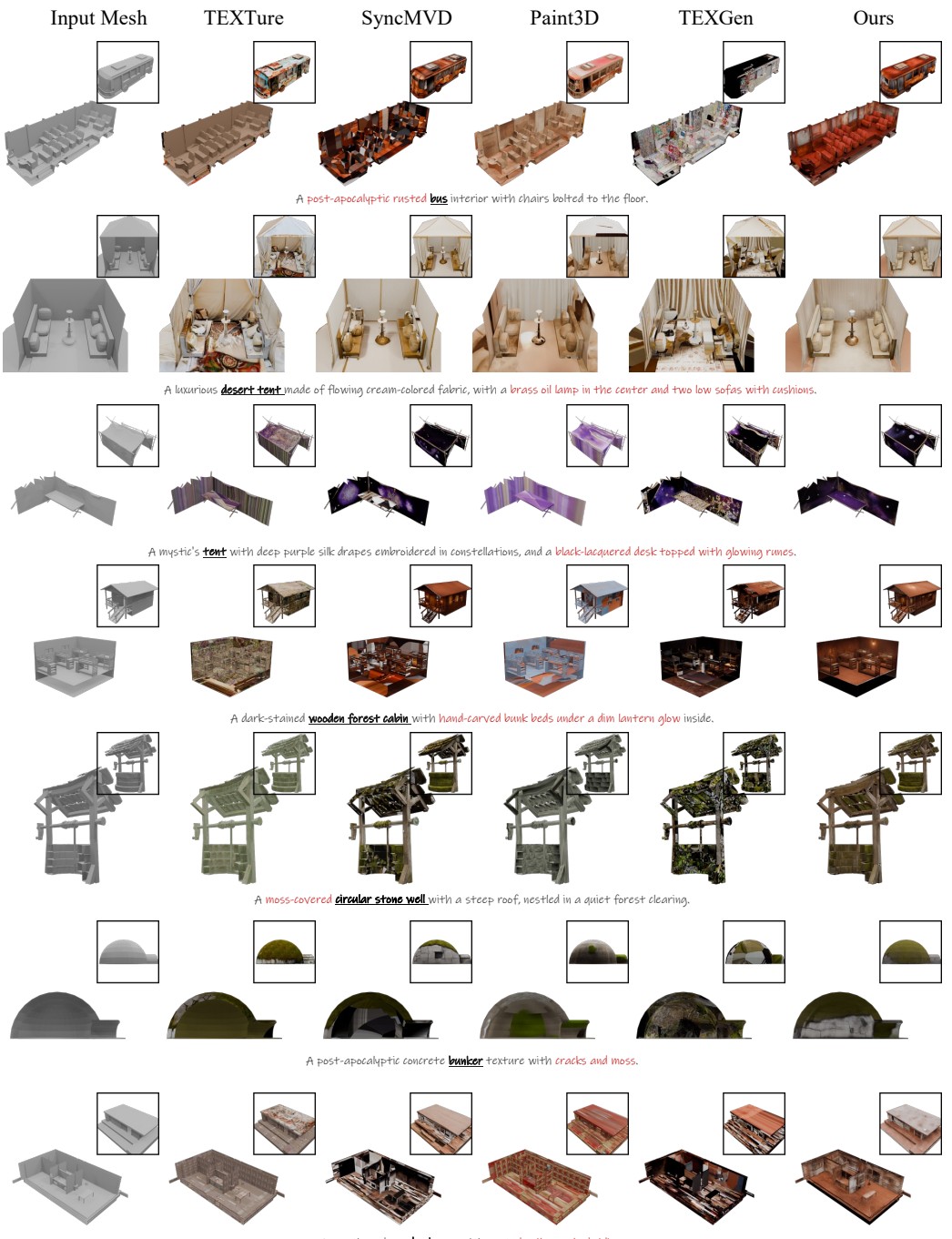

**Figure A17: Additional Qualitative Comparison.** You can view a video of the results and find more qualitative outcomes at this link.

A coral reef house with prismarine walls,
bubble columns, and glowing sea lanterns.

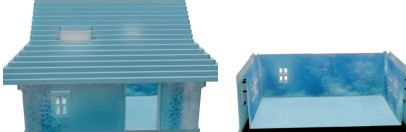

A fantasy-themed home theater with mural-painted walls,
canopy-covered seats, and carved runes lining the aisles.

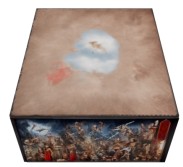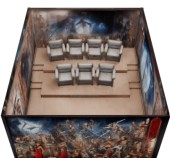

A painter's utility chest splattered with layered
pigment stains and lacquered over for texture.

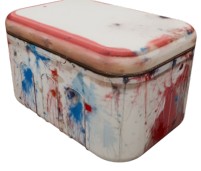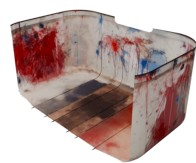

A bohemian artisan's toolbox painted in
layered mandala motifs and distressed edges.

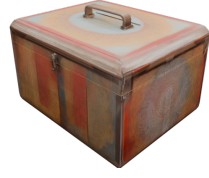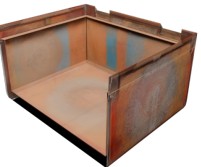

A marble-top dresser with polished stone
drawers and gilded trim along every edge.

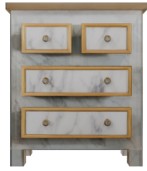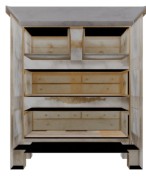

A gothic wardrobe with matte black wood,
silver inlaid filigree, and iron claw hinges.

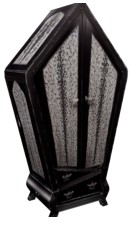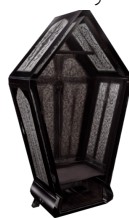

A lacquered oriental wardrobe with painted cranes
and wave motifs on the inner back panels.

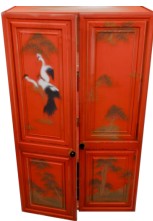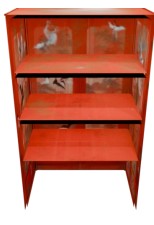

A fantasy lantern wrapped in silver branches,
with a blue-glowing enchanted candle at its core.

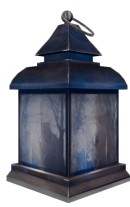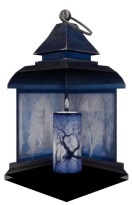

A classic car featuring wire-spoke wheels, quilted
leather interior, and brass instrument dials.

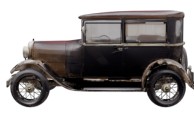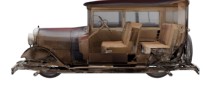

A lamp with stained glass panels arranged in floral
patterns and outlined in dark soldered seams.

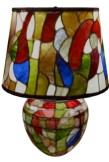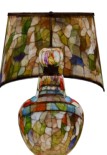

**Figure A18: Additional Qualitative Results.** You can view a video of the results and find more qualitative outcomes at this link.

