# OpenReview forum: "GOATex: Geometry & Occlusion-Aware Texturing"
_NeurIPS.cc/2025/Conference — NeurIPS 2025 poster_

### Official Review · Reviewer_8SvF · 2025-06-25

**Clarity:** 1
**Significance:** 3
**Originality:** 3
**Rating:** 5
**Confidence:** 4

**Summary:**

The paper proposes an approach to simultaneously generate the internal and external textures of 3D objects.The paper utilizes an atlas to segment independent faces into superfaces.It divides the superfaces into different levels based on the order of multiple ray hits on the superfaces.The paper proposes residual face cluster to maintain the complete structure for generating reasonable textures and normal flip to make more occluded areas visible.

**Questions:**

1.The hit level and its relationship with residual face cluster and normal flip are the core of your paper, which should be demonstrated more clearly rather than being placed in the supplement. This makes it difficult to follow.

2.The author has claimed to be the first work to simultaneously generate internal and external textures of objects, and the texture generation part relies on SyncMVD. Therefore, it is sufficient to compare the baseline SyncMVD and present more ablation studies. There is no need to compare with other texture generation models, as they completely collapse. Alternatively, such content can be moved to the supplement.

3.The ablation study uses complex car cases. Simple examples should also be presented to facilitate understanding of how different modules work.

4.Can a superface be assigned to different hit levels?

**Ethical Concerns:**

["NO or VERY MINOR ethics concerns only"]

**Final Justification:**

It has addressed most of my concerns, and I have decided to raise the score.

**Limitations:**

yes

**Quality:**

2

**Strengths And Weaknesses:**

Strengths：
The paper is the first to propose the work of simultaneously generating the internal and external textures of 3D objects.

Weaknesses：
The hit level is the core of the paper, and its changes under different settings should be described and presented in detail, rather than being included in the supplementary material.

---

> ### Author Rebuttal · Authors · 2025-07-31
>
> We thank the reviewer for recognizing the novelty of our work and for the constructive feedback on improving the clarity of our presentation.
>
> ---
>
> Q1: The hit level and its relationship with residual face cluster and normal flip are the core of your paper, which should be demonstrated more clearly rather than being placed in the supplement. This makes it difficult to follow.
>
> A1: We appreciate the feedback and will revise the main paper layout to more clearly present this pipeline in the core sections, rather than relying on the supplementary file.
>
>
> ---
>
> Q2.The author has claimed to be the first work to simultaneously generate internal and external textures of objects, and the texture generation part relies on SyncMVD. Therefore, it is sufficient to compare the baseline SyncMVD and present more ablation studies. There is no need to compare with other texture generation models, as they completely collapse. Alternatively, such content can be moved to the supplement.
>
> A2: Thank you for the constructive feedback. We agree that a more detailed comparison with SyncMVD, along with thorough ablation studies, is essential to highlight the core contributions of our occlusion-aware framework.
>
> To address this, we conducted additional experiments during the rebuttal period by constructing a series of ablated models, where components were cumulatively added on top of SyncMVD. This setup allows both step-by-step ablation and direct comparison with SyncMVD.
>
> For evaluation, we conducted A/B preference tests using SyncMVD as the reference, and each ablated variant as well as our full model as the comparison targets. To improve robustness, we employed multiple GPT-based evaluators (GPT-4o-mini, GPT-4o, GPT-4.1, and GPT-o3), and ran each configuration four times to account for response variability. In the table below, each score represents the win rate (%) of the corresponding method over SyncMVD.
>
> | Method | GPT-4o-mini | GPT-4o | GPT-4.1 | GPT-o3 | Average |
> |---|---|---|---|---|---|
> | + Hit Level Assignment | 82.50 | 66.67 | 75.00 | 77.50 | 75.68 |
> | + Superface Construction | 84.62 | 70.00 | 75.86 | 89.74 | 80.27 |
> | + Soft UV Merging | 79.49 | 82.05 | **90.00** | 95.00 | 86.49 |
> | + Residual Face Clustering | 77.50 | 72.50 | 88.89 | 84.84 | 81.17 |
> | + Normal Flipping & Backface Culling (**Ours**) | **86.84** | **92.31** | 86.67 | **97.50** | **91.16** |
>
> While we observe overall performance gains across stages for the advanced GPT models, a slight drop appears when the residual face clustering is applied without normal flipping and backface culling. This is expected, as residual clustering alone cannot fully resolve occlusion. In some cases, two geometrically adjacent faces may be grouped into the same hit level, even if one fully occludes the other from all external views. In such cases, the occluded face cannot be textured during its designated rendering stage, and once that stage completes, it is excluded along with the rest of the level, resulting in missed interior content.
>
> This limitation is mitigated in the final step by further applying normal flipping and backface culling, which introduces an indirect peeling-off effect. Instead of removing previously textured faces, we flip their normals and apply view-dependent culling—making them invisible from certain camera viewpoints. This increases the chance that previously occluded faces, which share the same hit level, are no longer blocked and can be exposed and textured in subsequent renderings. As a result, the method produces more complete and semantically coherent textures, especially in tightly occluded interior regions.
>
> It is also worth noting that while our final textured renderings show clear improvements in the complete pipeline, intermediate ablated methods may not always exhibit visually striking differences in the final outputs. This is because early-stage outputs often produce out-of-distribution (OOD) inputs to the depth-conditioned ControlNet, leading to suboptimal or unstable generations. As a result, the benefits of each component are most clearly observed in intermediate representations, such as depth maps and UV maps, that reveal differences in geometric completeness and structural coherence at each hit level. Therefore, to provide a comprehensive and intuitive understanding of each stage, we will also incorporate side-by-side comparisons of depth maps, UV layouts, and textured renderings in the revised ablation analysis.
>
>
> ---
>
> Q3.The ablation study uses complex car cases. Simple examples should also be presented to facilitate understanding of how different modules work.
>
> A3: As discussed in A2, we will incorporate a range of examples, from simple to complex, in the revised manuscript to more clearly illustrate the role of each module in our framework. These examples will be supported by step-by-step visualizations using depth maps, UV maps, and textured renderings, making the effect of each component easier to interpret. We believe this will enhance the clarity of the ablation study and help readers better understand how different modules contribute to the final results.
>
> ---
>
> Q4: Can a superface be assigned to different hit levels?
>
> A4: No, a superface is strictly assigned to a single hit level.
>
> Assigning a unique hit level to each superface allows our method to progressively reveal and texture occluded geometry in a controlled, layer-by-layer manner. In contrast, assigning multiple hit levels to a single superface would break this peeling-off progression, making it difficult to expose and texture occluded regions properly.

---

> > ### Comment · Reviewer_8SvF · 2025-08-06
> >
> > Thank you for your detailed rebuttal. It has addressed most of my concerns, and I have decided to raise the score.

---

### Official Review · Reviewer_KoBr · 2025-07-01

**Clarity:** 4
**Significance:** 3
**Originality:** 4
**Rating:** 5
**Confidence:** 5

**Summary:**

Current texture generative models overlook one important problem: generating textures also inside the objects. The proposed method is arguably the first to explicitly tackle this problem by proposing a combination of several clever strategies to decompose complex meshes and texture their interior parts. They introduce the following steps: a superface construction, a hit level assignment performed with ray casting, progressive visibility control and a per-level UV blending. None of the novel components need to be learned and are used in conjunction with an existing pre-trained texturing neural network (MVD). An extensive qualitative evaluation was paired with user and GPT-based studies.

**Questions:**

Besides mentioning the alternative techniques mentioned in the weakness section, I think the following questions need to be addressed:
1. The images used for evaluation (reported in A3) depicted 3 images representing the interiors and only one image representing the exterior. Considering that the proposed method is the only one specifically designed to texture interiors, I would consider the test slightly biased. I think that showing 2 images of the interiors and 2 of the exteriors or simply separating them would have been a fairer assessment.
2. The authors mention that other metrics like FID, KID, etc. are not well suited. I wonder if comparing against real or generated interior images could have been an option.
3. What is the number of participants recruited in the user study? I don't seem to find it either in the paper or in the supplementary materials.
4. It would be interesting to know where humans and GPT disagree and see some case where the other methods are better? Were the authors able to identify some patterns?
5. A potential limitation may be the computational overhead for the full pipeline, but this information is poorly reported. Authors mention in the appendix that the computational time depends on the number of faces, but do not provide estimates. I think that it would be nice to report a plot reporting the number of faces on the x-axis, and the computational time of all the proposed non-learning-based components on the y-axis. Additional information on the computational cost of MVD would also make the paper more self-contained.

Although I already lean towards acceptance, I would be extremely eager to further increase my score if these questions and the limitations were properly addressed.

**Ethical Concerns:**

["NO or VERY MINOR ethics concerns only"]

**Final Justification:**

The authors properly addressed all my questions. I am happy to increase my score.

**Limitations:**

I know that some limitations have been mentioned in the Supplementary materials, but they appear rather superficial. Every method has some limitations, and I would be interesting to hear the author’s opinion on some limitations of their proposed method. In the supplementary materials the main limitations are attributed to MVD, which is not the core contribution of this work. I would suggest being inspired by Section 2 of the paper checklist. Also, refer to Question 5 as a potential point of discussion.

**Paper Formatting Concerns:**

No major concerns but I wanted to flag that the instruction block of the checklist has not been deleted yet. Please, remember to do this before submitting the camera version ready of the paper (in case of positive outcome).

**Quality:**

3

**Strengths And Weaknesses:**

### Strengths
- The paper is generally well written and provides enough information to aid reproducibility
- The proposed method offers a non-learning-based solution that can be potentially leveraged by multiple image-based texture generators without any fine-tuning or training.
- Authors provided a very well-curated (anonymous) project page providing more results and animations. I am a strong believer that papers should be self-contained, therefore I consider it to be a nice addition which is not part of my assessment but still wanted to thank the authors for the additional effort.

### Weaknesses and Suggestions
- I think that it would be worth mentioning some methods that are not leveraging pre-trained image models and adopting different representations [i, ii, iii, iv]. These may be intrinsically capable of texturing objects inside, but in most cases given the more challenging generation settings (they do not operate on images), quality improvements would be needed for competing. Therefore, I think comparisons would be unnecessary. Yet mentioning alternative representations would make the related work and/or introduction more comprehensive.
- I have a few reservations for the Experiment Section, which are later detailed in the Questions section.
- I think that the limitations are not well analyzed. See Questions section.

### Minor Comments and Suggestions
- Adding \\( F_{k=0, ..., 3}^\text{init} \\) and \\( F_{k=0, ..., 3}^\text{res} \\) to Fig. 2 part 3 may aid the reader while reading Sec. 3.2. This addition may be part of the caption as well.
- Figure 3 should be ideally placed in Sec. 3.2 and Figure 4 in Sec. 3.3.
- In Figure 5, I believe the former name of the method (probably later renamed GOATex) is reported in all histograms.
- line 266: proressive -> progressive


PS. It would be a pity if the authors confirmed their decision not to release their code, it would make the paper considerably more impactful. This is obviously not part of the assessment, and it is perfectly understandable if this is not possible.


---------------

[i] Yu, X., Dai, P., Li, W., Ma, L., Liu, Z., & Qi, X. (2023). Texture generation on 3d meshes with point-uv diffusion. In Proceedings of the IEEE/CVF International Conference on Computer Vision (pp. 4206-4216).

[ii] Foti, S., Zafeiriou, S., & Birdal, T. (2024). Uv-free texture generation with denoising and geodesic heat diffusion. Advances in Neural Information Processing Systems, 37, 128053-128081.

[iii] Wei, J., Wang, H., Feng, J., Lin, G., & Yap, K. H. (2023). Taps3d: Text-guided 3d textured shape generation from pseudo supervision. In Proceedings of the IEEE/CVF conference on computer vision and pattern recognition (pp. 16805-16815).

[iv] Oechsle, M., Mescheder, L., Niemeyer, M., Strauss, T., & Geiger, A. (2019). Texture fields: Learning texture representations in function space. In Proceedings of the IEEE/CVF international conference on computer vision (pp. 4531-4540).

---

> ### Author Rebuttal · Authors · 2025-07-31
>
> We sincerely thank the reviewer for the detailed and constructive feedback. We believe the combination of precise editorial suggestions and thoughtful technical insights will significantly contribute to improving the clarity, depth, and completeness of the work.
>
> ---
>
> Q1: About the code release
>
> A1: Absolutely! We are happy to release the full codebase upon publication to support reproducibility and encourage further research.
>
> ---
>
> Q2: About alternative approaches that do not rely on pre-trained image models and use different representations.
>
> A2: Thank you for the suggestion. We agree that mentioning such alternative methods would make our related work more comprehensive. We will revise the manuscript to include a brief discussion of these representations in the Related Work section.
>
> ---
>
> Q3: Minor suggestions on clarifying notation in Fig. 2, repositioning Figures 3 and 4, correcting a method name in Fig. 5, and fixing a typo.
>
> A3: We sincerely appreciate your attentive and thoughtful comments. We will reflect suggested edits in the revision.
>
> ---
>
> Q4: The number of participants recruited in the user study.
>
> A4: The number of participants is reported in the supplementary material, Section A.3.1, line 58. For clarity, we summarize them here:
> + vs TEXTure: 21 participants
> + vs SyncMVD: 33 participants
> + vs Paint3D: 30 participants
> + vs TEXGen: 45 participants
>
> ---
>
> Q5: Conducting FID/KID-based evaluation by comparing against real or generated interior images.
>
> A5: Thank you for the suggestion.
>
> Following the reviewer’s recommendation, we conducted an FID-based evaluation using pseudo-reference images synthesized with depth-conditioned Stable Diffusion (SD 1.5), as the ground-truth meshes lack realistic interior textures. Notably, all evaluated methods are also based on SD 1.5.
>
> To generate pseudo-references, we rendered depth maps from the ground-truth meshes (per view and hit level) and used depth-conditioned SD 1.5 to synthesize reference images. These were used to compute FID scores against the corresponding outputs of each method, separately for exterior (hit level 1) and interior regions (hit levels ≥ 2). We averaged scores across views and meshes:
>
> | Method | FID (Exterior, ↓) | FID (Interior, ↓) |
> |---|---|---|
> | SyncMVD | **26.07** | **28.36** |
> | TEXTure | 26.23 | 28.65 |
> | Paint3D | 29.05 | 29.86 |
> | TEXGen | 30.74 | 31.21 |
> | Ours | 30.75 | 30.09 |
>
> While lower Interior FIDs of TEXTure and SyncMVD may appear favorable, they are misleading: (1) TEXTure leaves interiors blank or smooth, with occasional exterior texture “bleed” through mesh holes; SyncMVD extrapolates exterior colors via Voronoi heuristics. Neither produces semantically meaningful, prompt-aligned interiors. (2) The pseudo-references for hit levels ≥ 2, synthesized from single-view depth, lack multiview consistency and occlusion reasoning, often blending exterior and interior content, unlike MVD-based approaches in SyncMVD and ours.
>
> These low Interior FIDs largely reflect (1) exterior-like texture continuity in methods without actual interior synthesis, and (2) pseudo-references biased toward such appearances. In contrast, methods that generate semantically meaningful interiors may deviate from these references and be unfairly penalized. We will include this analysis and qualitative examples in the revised manuscript.
>
> Regarding exterior FID: Although our method scores higher, this is not a core limitation. GOATex specifically targets the challenging task of synthesizing textures for occluded interiors (hit-level ≥ 2), which prior work does not address. As GOATex is modular by design, high-fidelity exterior rendering (hit-level 1) can be integrated using existing methods. Our contribution thus complements, rather than replaces, current pipelines.
>
> ---
>
> Q6: Concern about potential bias in the qualitative evaluation due to imbalance between interior and exterior images.
>
> A6: Thank you for raising this important point.
>
> To address the concern regarding evaluation balance, we revised both the user study and GPT-based evaluation setups to ensure fair and unbiased comparisons between interior and exterior texturing. Specifically, following the visualization style used on our project page, we rendered for each method:
> + one GIF showing the exterior of the object, and
> + one GIF showing the interior via a cut-away or sliced view.
>
> These side-by-side visualizations were shown to participants or GPTs, providing one exterior and one interior view per method per object. Due to time constraints, we report results from 16 human raters who passed a vigilance test, with plans to expand this pool further. To compensate for this, we included four GPT variants, GPT-4o-mini, GPT-4o, GPT-4.1, and GPT-o3, and repeated each evaluation four times to reduce variance and improve robustness.
>
> Below are the aggregated preference rates (%) for GOATex over each baseline:
> | Baseline | Humans | GPT-4o-mini | GPT-4o | GPT-4.1 | GPT-o3 |
> |---|---|---|---|---|---|
> | TEXTure | 86.11 | 51.56 | 50.60 | 43.42 | 55.13 |
> | SyncMVD | 87.15 | 64.79 | 70.51 | 69.62 | 72.84 |
> | Paint3D | 83.33 | 85.51 | 83.13 | 82.72 | 73.75 |
> | TEXGen | 96.53 | 87.50 | 84.71 | 88.31 | 85.19 |
>
> While GOATex received strong and consistent preference from human raters across all comparisons, its relative advantage diminished when compared to TEXTure and SyncMVD in evaluations conducted with GPT-4 variants. We elaborate on this observation in A5.
>
> ---
>
> Q7: About disagreement patterns between human preference and GPT, and cases where other methods perform better.
>
> A7: Thank you for the interesting question.
>
> To better understand human-GPT disagreement patterns, we conducted both quantitative and qualitative analyses:
>
> 1. Correlation with human ratings (Pearson’s r; r = 1 indicates perfect correlation):
> + GPT-4o-mini: 0.22
> + GPT-4o: 0.31
> + GPT-4.1: 0.43
> + GPT-o3: 0.34
>
> 2. Agreement metrics (Cohen’s κ, averaged over κ > 0; κ = 1 indicates perfect agreement):
> + GPT-to-GPT: 0.54
> + User-to-user: 0.31
> + User-to-GPT: 0.27
>
> These quantitative results indicate that:
> + Advanced GPT models (especially GPT-4.1 and GPT-o3) show stronger alignment with human raters.
> + GPT models exhibit higher internal consistency than human raters.
> + Notably, in 17 cases, user-to-GPT agreement exceeded κ = 0.5, and in two cases, GPT-o3 achieved perfect agreement (κ = 1.0) with a human rater, demonstrating that GPTs can align closely with human preferences in specific contexts.
>
> 3. Qualitative analysis further revealed a consistent pattern of divergence:
> + Humans weighed both interior and exterior quality and particularly penalized incomplete or semantically incoherent or incomplete interiors.
> + GPTs tended to favor sharper exterior textures, even when interiors were missing or incorrect — a bias consistent with FID-based evaluation discussed in A5.
>
> This explains why GPTs often underrated our method: GOATex is designed for occluded interiors, whose quality humans appreciate more than GPTs, which emphasize visible exterior realism. We will include representative examples and full results in the revised manuscript to better contextualize these differences.
>
> ---
>
> Q8: About the computational cost of the full pipeline.
>
> A8: Thank you for the insightful suggestion.
>
> We systematically analyzed the runtime of our pipeline across meshes with increasing face counts by progressively subdividing mesh faces of five representative assets. For each mesh, we measured the number of faces and superfaces, time spent on superface construction, hit-level assignment, and MVD-based rendering & synthesis. The table below shows the average across the assets:
>
> | # Faces | # Superfaces | Superface Construction (s) | Hit-Level Assignment* (s) | Rendering & Texturing (s) |
> |---|---|---|---|---|
> | 5k | 196.0 | 0.23 | 259.57 | 117.93 |
> | 10k | 221.2 | 0.46 | 295.38 | 121.80 |
> | 20k | 247.6 | 0.86 | 314.16 | 130.42 |
> | 40k | 433.0 | 1.87 | 394.67 | 146.45 |
> | 80k | 586.0 | 3.99 | 539.68 | 197.29 |
> | 160k | 2033.2 | 18.59 | 942.36 | 549.83 |
> | 320k | 5418.0 | 57.76 | 2563.35 | 2769.22 |
>
> (*Hit-level assignment is reusable when generating multiple variants.)
>
> As the reviewer correctly pointed out, the number of mesh faces has a significant impact on runtime. In our Objaverse 1.0-based dataset, however, mesh sizes are typically modest (median: 8.5k faces), keeping total runtime practical (under ~10 minutes in most cases).
>
> We also find that the number of superfaces grows sublinearly with mesh size. While the pipeline is designed to operate at the superface level, most steps currently rely on face-level processing due to the reliance of existing implementations. Moving to superface-level processing throughout is a promising direction for reducing runtime on high-res meshes.
>
> We will include this analysis and a corresponding runtime plot in the revision.
>
> ---
>
> Q9: Request for a deeper discussion of the limitations of the proposed method beyond what's noted in the submitted version.
>
> A9: Thank you for the insightful question.
>
> One key limitation of GOATex lies in its semantically inconsistent hit-level assignment. Our current decomposition relies solely on geometric visibility computed via multi-view ray casting, which can result in semantically coherent regions being split across different hit levels (as also noted by Reviewer nix7 in Q4). While this does not lead to noticeable seams or texture artifacts in our experiments, it highlights a structural limitation of relying purely on geometry. Incorporating semantic-aware grouping into the hit-level assignment could enable more fine-grained texturing by allowing the pipeline to operate at the level of semantic entities. This would make it possible to assign different prompts to distinct parts, opening the door to richer, more controllable synthesis beyond the current progressive outside-in approach.
>
> We will include this discussion in the revised manuscript to clarify the limitation and motivate future extensions.

---

> > ### Comment · Reviewer_KoBr · 2025-08-04
> >
> > I am very pleased with the authors' response. The additional experiments and promised amendments will be a nice addition to the paper.
> >
> > I do believe that FID and KID scores are not necessarily indicators of poor performance of the proposed method, but I would still encourage authors to add images supporting these results.
> >
> > Similarly, the table in A8 would be more readable as a plot (which was not allowed as part of this rebuttal).
> >
> > I thank the authors for their efforts and will increase my score accordingly.

---

### Official Review · Reviewer_nix7 · 2025-07-01

**Clarity:** 3
**Significance:** 3
**Originality:** 4
**Rating:** 5
**Confidence:** 4

**Summary:**

The paper introduces GOATex, a diffusion-based framework for 3D mesh texturing that addresses the challenge of generating realistic textures for both exterior and occluded interior surfaces. Existing methods struggle with interior texturing due to limited access to occluded geometry, often resulting in incomplete textures and visible seams. GOATex leverages ray-based visibility analysis to decompose the mesh into ordered layers, enabling progressive texturing from the outermost to innermost regions. Through extensive experiments, GOATex outperforms existing baselines, generating high-quality and seamless textures for both visible and occluded surfaces of untextured objects.

**Questions:**

When dealing with an object that is not entirely closed, such as a vase with a small opening at the top, how should we assign the hit level? Specifically, does the inner bottom of this vase belong to Hit Level 1, while other inner surfaces belong to Hit Level 2? In such scenarios, a critical challenge arises: how to ensure that the texture of the inner bottom aligns with the textures of other inner surfaces that belong to Hit Level 2, thereby maintaining a consistent and coherent visual appearance?

**Ethical Concerns:**

["NO or VERY MINOR ethics concerns only"]

**Limitations:**

As elucidated in the Weaknesses section, GOATex may face significant difficulties when dealing with substantially more complex 3D meshes.

**Quality:**

3

**Strengths And Weaknesses:**

**Strengths**
1. The authors pose a thought-provoking question, namely, generating realistic textures for occluded surfaces of 3D meshes, and bring this issue to the forefront of academic scrutiny.
2. Novel and effective Occlusion-Aware Approach. The proposed Occlusion-Aware Approach, termed GOATex, introduces a novel and effective paradigm for texturing 3D meshes, employing a ray-based and layer-by-layer strategy that progresses from the outermost to innermost surfaces.
3. The comprehensive evaluation, including a user study, demonstrates the efficacy of this method in generating realistic textures for both visible and occluded surfaces

**Weaknesses**
1. The approach struggles to effectively handle complex 3D meshes comprising multiple layers.
2. A notable limitation of the method is that certain occluded areas on interior surfaces are neglected, as exemplified by the back of chairs on the bus shown in Figure 2.
3. The framework lacks a clear mechanism for adaptively adjusting the corresponding text prompt in response to varying hit levels.

---

> ### Author Rebuttal · Authors · 2025-07-31
>
> We sincerely thank the reviewer for recognizing the importance and novelty of the proposed task. We appreciate the encouraging feedback on our occlusion-aware approach and the overall evaluation.
>
> ---
>
> Q1: The approach struggles to effectively handle complex 3D meshes comprising multiple layers.
>
> A1: We thank the reviewer for raising this point.
>
> Scene complexity arises from various factors, including the number of semantic entities, their spatial arrangement, and the underlying geometry. As the number of semantic entities increases, the mesh tends to become more layered and entangled, making it increasingly difficult to disambiguate structures and generate proper textures across them. Furthermore, a high face count also complicates UV mapping, which can negatively affect texture quality and consistency.
>
> In our current setup, the evaluated meshes span a broad range of semantic and geometric complexity, from simple to highly intricate cases. While it is not straightforward to define or quantify the number of "layers" in a scene, each mesh contains on average about 1.35 semantic entities, with some containing up to 7. The geometric complexity also varies significantly, with a median face count of 8.5K and a maximum of 166K.
>
> Qualitatively, our method is demonstrated to perform well even on meshes with such multiple internal structures and layers. For instance, in the car example shown in Figure 6 of the main paper, components such as the steering wheel and seats are relatively small yet semantically important. While these parts are challenging to isolate and texture individually, our method (GOATex) is able to produce plausible textures, whereas baseline methods fail to generate meaningful results in such settings. Additionally, as illustrated in row 3, column 1; row 6, column 1; and row 6, column 3 of our project page, our approach generates coherent and prompt-aligned textures even in scenes with multiple internal or layered components. This is further supported by our quantitative evaluation, where our method is strongly preferred over baselines by both human raters and GPT-based assessments.
>
> That being said, we acknowledge that even more compositionally rich and deeply nested configurations could be imagined. For example, consider a children's bedroom scene containing a closed toy chest filled with plush dolls, a bunk bed with built-in drawers and under-bed compartments, and a ride-on toy car with fully modeled interior components. Such scenes involve numerous interleaved objects with varied spatial and semantic relationships, leading to extreme geometric and semantic complexity.
>
> However, due to this practical consideration, we chose not to focus our evaluation on such a setup, as texturing highly nested and composite scenes all at once is rare in typical workflows. In practice, these environments are more effectively and commonly constructed by texturing individual components with moderate levels of hidden internal structures (e.g., a closed toy chest, plush dolls, a bunk bed with built-in drawers, fully enclosed ride-on toy cars) separately, and then combining them into a larger scene graph or hierarchical assembly. Automating this compositional process remains a promising direction for future work.
>
> ---
>
> Q2: A notable limitation of the method is that certain occluded areas on interior surfaces are neglected, as exemplified by the back of chairs on the bus shown in Figure 2.
>
> A2: We thank the reviewer for the comment. We believe the concern may stem from the specific view shown in Figure 2 in the main paper, where only the front of the chairs is visible, and the backs are not shown from that particular camera angle. However, we would like to clarify that in our method, each mesh face is assigned a hit level based on multi-view ray casting from a densely sampled hemispherical camera rig (as mentioned in Section 4.1, L195). This setup ensures that all visible and occluded surfaces (including the backs of interior objects) are intersected by at least one ray and are thus included in the progressive texturing process.
>
> ---
>
> Q3: The framework lacks a clear mechanism for adaptively adjusting the corresponding text prompt in response to varying hit levels.
>
> A3: Thank you for bringing this up. Upon reviewing our submission, we acknowledge that we did not clearly describe how dual prompting is enabled within our framework and we appreciate the opportunity to clarify it.
>
> As described in Section 3 (L172–174), we generate a texture for each hit level by rendering multi-view depth maps of the geometry corresponding to that level and synthesizing textures using a text-guided MVD module. Then, as noted in L178, the resulting textures from each hit level are merged into a unified UV texture map via visibility-weighted blending.
>
> This design naturally enables layer-specific prompting: since the MVD module is conditioned on depth maps and a textual prompt independently per hit level, the prompt used at each level can be chosen independently. In practice, we assign a distinct prompt to the outermost layer (hit-level = 1) to control the exterior appearance, and a different prompt for hit levels ≥ 2 to control interior textures. This mechanism supports what we call dual prompting, as shown qualitatively in Figures 1 and 7.
>
> We will make this sufficiently explicit in the revised version of the manuscript.
>
> ---
>
> Q4: Potential issues in maintaining texture coherence when faces within a semantically coherent region are assigned to different hit levels
>
> A4: Thank you for the thoughtful question.
>
> As the reviewer correctly points out, our current hit level assignment is purely geometric, based on ray intersection depth, and therefore agnostic to semantic information. In certain edge cases, such as a vase with a narrow opening, semantically coherent interior regions (e.g., the connected inner surface) may be assigned to different hit levels due to partial visibility from external viewpoints. This could, in principle, result in textural discontinuities across those regions.
>
> However, in practice, we empirically observe that our method still produces semantically plausible and coherent textures. We attribute this to several key factors:
> 1. Residual face clustering, combined with view-dependent normal flipping and backface culling (Section 3.2), preserves structural continuity across hit levels in the rendered depth maps. As shown in Figure 3 (right), although each hit level targets a distinct subset of faces (particularly those previously occluded), our method consistently maintains the overall geometry, retaining a globally coherent silhouette and shape in the depth maps. This inter-level consistency ensures that the diffusion model receives a stable geometric context at each synthesis stage, even though the specific regions being textured differ. As a result, as shown in the rendered textures in Figure 3 (right), semantic coherence is naturally preserved across hit levels.
> 2. In addition, our soft UV-space blending strategy (Section 3.3) further reinforces this continuity by smoothly interpolating textures across hit levels using view-dependent visibility weights. This weighted blending mitigates hard seams and ensures stylistic consistency in regions where semantic surfaces span multiple hit levels, as can be seen in Figure 4.
>
> Together, these two components work in concert to ensure that our method produces coherent and semantically aligned textures, even in cases where semantically unified regions are geometrically split across hit levels.
>
> That said, we acknowledge this limitation and view it as a promising direction for future work. Specifically, incorporating semantic-aware refinement of hit levels, for example by further grouping superfaces that belong to the same semantic volume with pre-trained part segmentation modules [*], could further improve consistency in such edge cases.
>
> Additionally, while not explored in our current implementation due to computational resource constraints, an even more integrated strategy could involve performing soft UV-space blending during each denoising process itself, rather than after hit-level-specific texture synthesis. This would enable simultaneous multi-view and multi-hit-level texturing, potentially allowing for tighter coherence across depth layers.
>
> We thank the reviewer for highlighting this nuanced and important point, and will clarify this limitation and potential mitigation strategies in the revision.
>
> [*] Wu et al., Point Transformer V3: Simpler, Faster, Stronger, In CVPR, 2024

---

> > ### Comment · Reviewer_nix7 · 2025-08-05
> >
> > I thank the authors for their detailed response, which has addressed most of my concerns. I recommend that the discussion regarding the model's performance on scene complexity and how to ensure semantically coherent textures be integrated into the revision, which I believe will strengthen the paper.

---

> > > ### Author Response · Authors · 2025-08-05
> > >
> > > We sincerely appreciate the reviewer’s feedback and will incorporate the discussion on scene complexity and semantic coherence into the revised version.

---

### Official Review · Reviewer_imB6 · 2025-07-06

**Clarity:** 4
**Significance:** 3
**Originality:** 4
**Rating:** 5
**Confidence:** 4

**Summary:**

This paper presents a diffusion-based method for texturing shapes that contain both exterior and interior surfaces. Prior work on diffusion-based shape texturing typically renders a camera view, making use of a depth conditioned multi-view diffusion model to generate uv-textures. This work points out that such a pipeline will generally render (and hence texture) only exterior surfaces, but will fail to texture the interior of a car or building shape. The key architectural innovation is to segment a 3D shape into an "onion-layered" collection of surfaces ordered from outside-in. Such segmented shapes can be used to generate segmented depth maps, which can then be combined with depth-conditioned multi-view diffusion models to generate outside-in ordered uv-texture maps (which are then be blended back to obtain a fully inside-out textured model).

**Questions:**

Please answer the first "weakness" above, regarding volumetric texture models.

**Ethical Concerns:**

["NO or VERY MINOR ethics concerns only"]

**Final Justification:**

This is a nice submission.

**Limitations:**

yes

**Quality:**

4

**Strengths And Weaknesses:**

Strengths

- The problem formulation is quite creative and the execution seems solid. The rendered visuals are simply fun to look at!
- This paper includes a link to a github project page, which may or maynot be allowed under the submission guidelines. I personally found the visuals quite helpful.

Weaknesses
- It would be interesting to compare to NERF-like approaches that learn a full volumetric texture model rather than a uv-based texture model. Such texturing pipelines can naturally model interior surfaces simply by rendering an interior camera, or even a "cut-away" view such as those shown in the final visuals. Examples of such volumetric (and not uv-based) texturing approaches include FlashTex and Fantasia3D. Could such methods be more readily adapted to modeling interior surfaces?
- Minor gripe: I find the term "occlusion-aware texturing" to be somewhat ill-defined, since occlusion is associated with a camera viewpoint but shapes are full 3D. One could use it in the sense of view-dependant (specular) materials, but there already exists a large body of work in this vain. I think the more proper term is texturing of models that allow for diverse camera viewpoints, including interior cameras and "cut-away" cameras.

---

> ### Author Rebuttal · Authors · 2025-07-31
>
> We sincerely thank the reviewer for their positive and encouraging feedback. We’re glad that the problem formulation and execution were found to be creative and solid, and that the rendered visuals were enjoyable and helpful.
>
> ---
>
> Q1: It would be interesting to compare to NERF-like approaches that learn a full volumetric texture model rather than a uv-based texture model.
>
> A1: We appreciate the reviewer’s suggestion to consider NeRF-like or volumetric texturing approaches such as FlashTex and Fantasia3D. As noted, in principle, these methods can support interior or cut-away views and can be combined with diffusion-based supervision (e.g., SDS loss) to optimize radiance fields for interior volumes or regions.
>
> However, applying such methods directly without additional geometric and occlusion-aware constraints can be problematic. In practice, when training is performed primarily from a set of fixed external views, NeRF-like models tend to accumulate high density at the outermost surfaces. This effectively blocks camera rays from reaching the interior, preventing supervision signals from propagating to inner geometry. As a result, color radiance in these regions often remains under-constrained or unlearned.
>
> One might attempt to mitigate this by gradually moving cameras inward or narrowing the frustum to expose internal volumes. However, without access to mesh structure, there is no guidance on how to move the cameras or which regions of the volume to expose. This often leads to indiscriminate optimization over the entire volume, introducing spurious geometry and severely degrading training efficiency. Even worse, NeRF-like models may recursively form nested high-density pseudo-surfaces, i.e. much like a troika structure, resulting in a degenerate, continuous shell-like volume that fails to capture meaningful interior surfaces.
>
> On the other hand, if mesh structure is available, it becomes possible in principle to expose and supervise surface layers in a controlled, outside-in manner. But doing so effectively still requires an explicit mechanism to identify occluded geometry at each depth and to organize supervision accordingly. This is precisely the functionality that our method introduces through hit-level assignment and visibility-depth analysis.
>
> Thus, we would like to emphasize that the core challenge in interior surface texturing is not the choice of representation (e.g., UV-based vs. volumetric), but the ability to perform geometry- and occlusion-aware visibility control. Once such control is established, the decision to optimize a UV texture map or a volumetric radiance field becomes secondary. Our framework provides the foundational mechanism required for either setting, and could be integrated into volumetric pipelines as a principled method for layered, surface-aware supervision, particularly in the presence of occluded or complex internal geometry.
>
> ---
>
> Q2: Potential misleading use of "occlusion-aware texturing"
>
> A2: We thank the reviewer for pointing out the potential ambiguity in the term “occlusion-aware.”
>
> While occlusion is often treated as a view-dependent concept, it can also be view-invariant in the case of structurally enclosed interior regions. For instance, exterior surfaces may or may not be visible depending on the viewpoint, but interior surfaces that are fully enclosed by outer geometry remain occluded from all external views, regardless of camera position.
>
> GOATex is specifically designed to address such occlusions. Our method explicitly identifies these deeply embedded, structurally occluded regions (i.e., faces that are never visible in standard multi-view setups) and progressively reveals them via ray-based visibility analysis. By decomposing the mesh into ordered hit levels and applying layer-wise texturing with blending, GOATex provides a geometric, structure-aware approach to handling occlusion that complements and extends conventional view-aware techniques.
>
> In that sense, we believe the term “occlusion-aware” is appropriate, as our method does not rely solely on view-based visibility cues, but instead explicitly identifies and handles regions that are consistently hidden due to the mesh’s geometric enclosure. That said, we understand the concern and are open to adopting alternative terminology in future revisions to avoid confusion with view-dependent material modeling.

---

### Decision · Program_Chairs · 2025-09-17

**Decision:**

Accept (poster)

**Comment:**

This submission received four "Accept" scores from the reviewers. The reviewers appreciated novelty of the contributions, thoroughness of the evaluation setup, as well as clear presentation and additional results. The remaining questions were addressed in the rebuttal. The final recommendation is therefore to accept.